# Qualitative similarities and differences in visual object representations between brains and deep networks

Georgin Jacob[1,2], R. T. Pramod[1,2], Harish Katti[1] & S. P. Arun [1✉]

Deep neural networks have revolutionized computer vision, and their object representations across layers match coarsely with visual cortical areas in the brain. However, whether these representations exhibit qualitative patterns seen in human perception or brain representations remains unresolved. Here, we recast well-known perceptual and neural phenomena in terms of distance comparisons, and ask whether they are present in feedforward deep neural networks trained for object recognition. Some phenomena were present in randomly initialized networks, such as the global advantage effect, sparseness, and relative size. Many others were present after object recognition training, such as the Thatcher effect, mirror confusion, Weber's law, relative size, multiple object normalization and correlated sparseness. Yet other phenomena were absent in trained networks, such as 3D shape processing, surface invariance, occlusion, natural parts and the global advantage. These findings indicate sufficient conditions for the emergence of these phenomena in brains and deep networks, and offer clues to the properties that could be incorporated to improve deep networks.

[1] Centre for Neuroscience, Indian Institute of Science, Bangalore, India. [2] Department of Electrical Communication Engineering, Indian Institute of Science, Bangalore, India. ✉email: sparun@iisc.ac.in

*How do I know this is true?*
*I look inside myself and see.*

*Tao Te Ching*[1]

Convolutional or deep neural networks have revolutionized computer vision with their human-like accuracy on object-recognition tasks, and their object representations match coarsely with the brain[2,3]. Yet they are still outperformed by humans[4,5] and show systematic finer-scale deviations from human perception[6–9]. Even these differences are largely quantitative in that there are no explicit or emergent properties that are present in humans but absent in deep networks. It is possible that these differences can be fixed by training deep networks on larger datasets, incorporating more constraints[3] or by modifying network architecture such as by including recurrence[10,11].

Alternatively, there could be substantive qualitative differences between how brains and deep networks represent visual information. This is an important question because resolving qualitative differences might require non-trivial changes in network training or architecture. A naïve approach would be to train deep networks on multiple visual tasks and compare them with humans, but the answer would be insightful only if networks fail to learn certain tasks[12]. A more fruitful approach would be to compare qualitative or emergent properties of our perception with that of deep networks without explicitly training them to show these properties.

Fortunately, many classic findings from visual psychology and neuroscience report emergent phenomena and properties that can be directly tested on deep networks. Consider for instance, the classic Thatcher effect (Fig. 1a), in which a face with 180° rotated parts looks grotesque in an upright orientation but looks entirely normal when inverted[13]. This effect can be recast as a statement about the underlying face representation: in perceptual space, the distance between the normal and Thatcherized face is presumably larger when they are upright than when they are inverted (Fig. 1b). Indeed, this has been confirmed using dissimilarity ratings in humans[14]. These distances can be compared for any representation, including for a deep network (Fig. 1c). Since deep networks are organized layer-wise with increasing complexity across layers, measuring these distances would reveal the layers at which the deep network begins to experience or "see" a Thatcher effect (Fig. 1d).

Knowing whether deep networks exhibit such effects can be insightful for several reasons. First, it would tell us whether the deep network indeed does "see" the effect the way we do. Second, this question can be asked of any deep network without explicit training to produce this effect. For instance, testing this question on network trained on object and face classification would reveal which kind of training is sufficient to produce the Thatcher effect. Finally, this question has relevance to neuroscience, because object representations in the early and late layers of deep networks match with early and late visual processing stages in the brain[10,15]. The layer at which this effect arises could therefore reveal its underlying computational complexity and offer clues as to its possible neural substrates.

Here, we identified as many emergent perceptual and neural properties as possible from visual psychology and neuroscience that can be recast as statements about distances between images in the underlying perceptual/neural representation. To investigate which properties arise due to architecture versus training, we tested each property on both randomly initialized deep networks as well as state-of-the-art deep networks trained on object classification tasks. This revealed some properties to be present in randomly initialized networks, some to be present only after object classification training and yet others that are absent after training. Such characterizations are an important first step in our understanding particularly in the

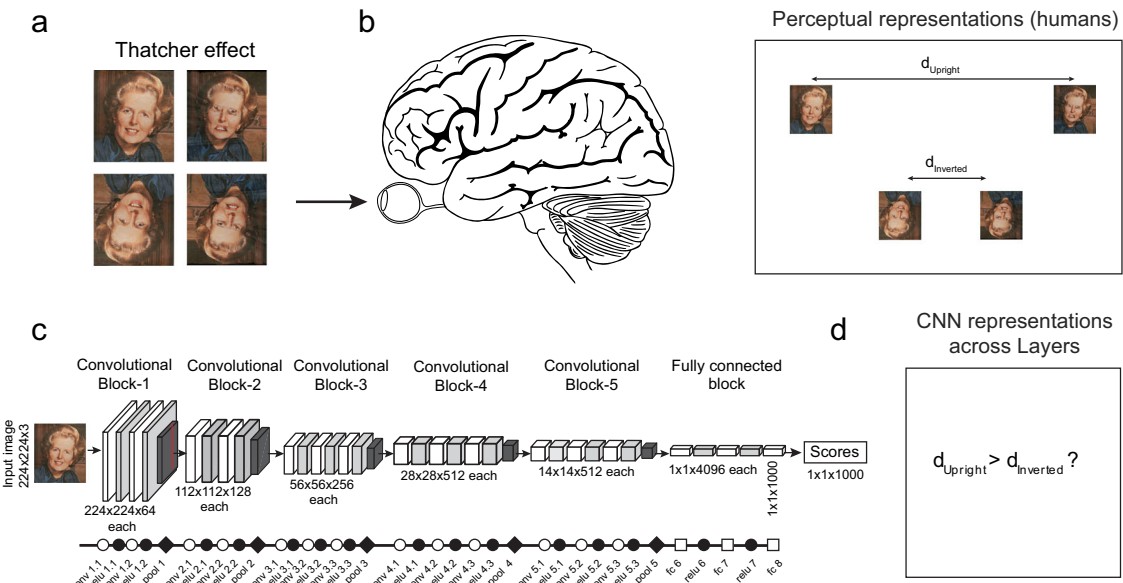

**Fig. 1 Evaluating whether deep networks see the way we do. a** In the classic Thatcher effect, when the parts of a face are individually inverted, the face appears grotesque when upright (*top row*) but not when inverted (*bottom row*). Figure credit: Reproduced with permission from Peter Thompson. **b** When the brain views these images, it presumably extracts specific features from each face so as to give rise to this effect. We can use this idea to recast the Thatcher effect as a statement about the underlying perceptual space. The distance between the normal and Thatcherized face is larger when they are upright compared to when the faces are inverted. This property can easily be checked for any computational model. *Brain Image credit: Wikimedia Commons.* **c** Architecture of a common deep neural network (VGG-16). Symbols used here and in all subsequent figures indicate the underlying mathematical operations perfomed in that layer: *unfilled circle* for convolution, *filled circle* for ReLu, *diamond* for maxpooling and *unfilled square* for fully connected layers. Unfilled symbols depict linear operations and filled symbols depict non-linear operations. **d** By comparing the distance between upright and inverted Thatcherized faces, we can ask whether any given layer of the deep network sees a Thatcher effect.

absence of useful theoretical accounts of both vision and deep networks.

## Results

We identified as many emergent perceptual and neural properties as possible that could be tested in deep networks without explicitly training them on tasks. We organized these properties broadly into five groups: (1) those that pertain to object or scene statistics, namely the Thatcher effect, mirror confusion and object–scene incongruence; (2) those that pertain to tuning properties of neurons in visual cortex, namely multiple object tuning and correlated sparseness; (3) those that pertain to relations between object features, namely Weber's law, relative size and surface invariance; (4) those that pertain to 3D shape and occlusions and (5) those that pertain to object parts and global structure.

For ease of exposition, we report below the results from a state-of-the-art pre-trained feedforward convolutional network, VGG-16, that is optimized for object classification on a large-scale image database (ImageNet), which contains 1.2 million training images across 1000 categories[16]. We also report the results of a randomly initialized VGG-16 network. This serves as a control to confirm that the property being investigated is due to training and not solely due to the architecture. We obtained similar results for another instance of the VGG-16 network trained with a different random seed (Supplementary Section S1), and similar results using several other feedforward deep networks varying widely in their depth and architecture (Supplementary Section S2). Likewise, we report all results here using Euclidean distance but obtained similar results for other distance metrics (Supplementary Section S3). Thus all our findings are generally true across a variety of network architectures as well as distance metrics but are specific to networks trained for visual object classification.

For each property, we performed an experiment in which we used carefully controlled sets of images as input to the network (in most cases , the same images as used in the behavioural/neural experiments). For these images, we obtained the activations of the units in each layer, and asked whether each layer shows that property. For simplicity, we deemed a network to show a property if it is present in the last fully connected layer, since the activations in this layer are used for eventual object classification. Interestingly, in some cases, the property temporarily emerged in intermediate or fully connected layers and vanished in the final layer, suggesting that it may be required for intermediate level computations but not for eventual classification.

**Experiment 1: Do deep networks see a Thatcher effect?** The Thatcher effect is an elegant demonstration of how upright faces are processed differently from inverted faces, presumably because we encounter mostly upright faces. As detailed earlier, it can be recast as a statement about the underlying distances in perceptual space: that normal vs. Thatcherized faces are closer when inverted than when upright (Fig. 2a). For each layer of the deep network (VGG-16), we calculated a "Thatcher index" of the form $(d_{upright} - d_{inverted})/(d_{upright} + d_{inverted})$, where $d_{upright}$ is the distance between normal and Thatcherized face in the upright orientation, and $d_{inverted}$ is the distance between them in an inverted orientation. Note that the Thatcher index for a pixel-like representation (where the activation of each unit is proportional to the brightness of each pixel in the image) will be zero since $d_{upright}$ and $d_{inverted}$ will be equal. For human perception, since $d_{upright} > d_{inverted}$, the Thatcher index will be positive. We estimated perceptual dissimilarities from a previous study that reported similarity scores between upright and inverted faces[14].

We calculated the Thatcher index across layers for three networks with similar architecture but differing in training (see

the "Methods" section). The first was VGG-16 which is trained for object classification[16]. The second was VGG-face which is trained for face recognition[17]. The third one was a randomly initialized VGG-16 network (VGG-16-rand). The Thatcher index for all three networks across layers is shown in Fig. 2b. It can be seen that the VGG-16 shows a positive Thatcher index in two convolutional layers (conv4 and conv5) and in the initial fully connected layers but eventually showed a weak negative effect in the final two fully connected layers (Fig. 2b). However, the randomly initialized VGG-16 network showed no such effect (Fig. 2b). By contrast, the VGG-face network showed a steadily rising Thatcher effect across layers that remained high in the fully connected layers (Fig. 2b). Thus, the Thatcher effect is strongest in deep networks trained on upright face recognition, weakly/temporarily present in networks trained on object recognition and entirely absent in a randomly initialized network.

**Experiment 2: Do deep networks show mirror confusion?** Mirror reflections along the vertical axis appear more similar to us than reflections along the horizontal axis (Fig. 2c). This effect has been observed both in behaviour as well as in high-level visual cortex in monkeys[18]. To assess whether deep networks show mirror confusion, we calculated a mirror confusion index of the form $(d_{horizontal} - d_{vertical})/(d_{horizontal} + d_{vertical})$, where $d_{horizontal}$ and $d_{vertical}$ represent the distance between horizontal mirror image pairs and between vertical mirror image pairs, respectively (see the "Methods" section). Since vertical mirror images are more similar in neurons, this index is positive (Fig. 2d). Across the deep network VGG-16, we found an increasing mirror confusion index across layers (Fig. 2d). This trend was absent in the randomly initialized network (VGG-16 rand; Fig. 2d). Thus, like human perception, deep networks also show stronger mirror confusion for vertical compared to horizontal mirror images.

**Experiment 3: Do deep networks show scene incongruence?** Our ability to recognize an object is hampered when it is placed in an incongruent context[19,20], suggesting that our perception is sensitive to the statistical regularities of objects co-occurring in specific scene context. To explore whether deep networks are also sensitive to scene context, we gave as input the same images tested on humans, and asked how the deep network classification output changes with scene context (see the "Methods" section).

An example object (hatchet) placed against a congruent context (forest) and incongruent context (supermarket) are shown in Fig. 2e. The VGG-16 network returned a high probability score in the congruent context (Fig. 2e, top row) but gave a low probability score for the same object in an incongruent context (Fig. 2e bottom row). We obtained similar results across all congruent/incongruent scene pairs: the VGG-16 top-1 accuracy dropped substantially for incongruent compared to congruent contexts (drop in accuracy from congruent to incongruent scenes: 20% for top-5 accuracy; 27% for top-1 accuracy; Fig. 2f). On the same scenes, human object naming accuracy also dropped for incongruent scenes, but the drop was smaller compared to the VGG-16 network (drop in human accuracy from congruent to incongruent scenes: 14% in the Davenport and Potter, 2004; 13% in Munneke et al. 2013; Fig. 2g). We note that assessing the statistical significance of the accuracy difference in humans and deep networks is not straightforward since the variation in accuracy reported are across subjects for humans and across scenes for the VGG-16 network.

The above results are based on comparing classification accuracy since that is the data available for scene incongruence in humans. To characterize the progression of the scene incongruence effect, we calculated the distance between each

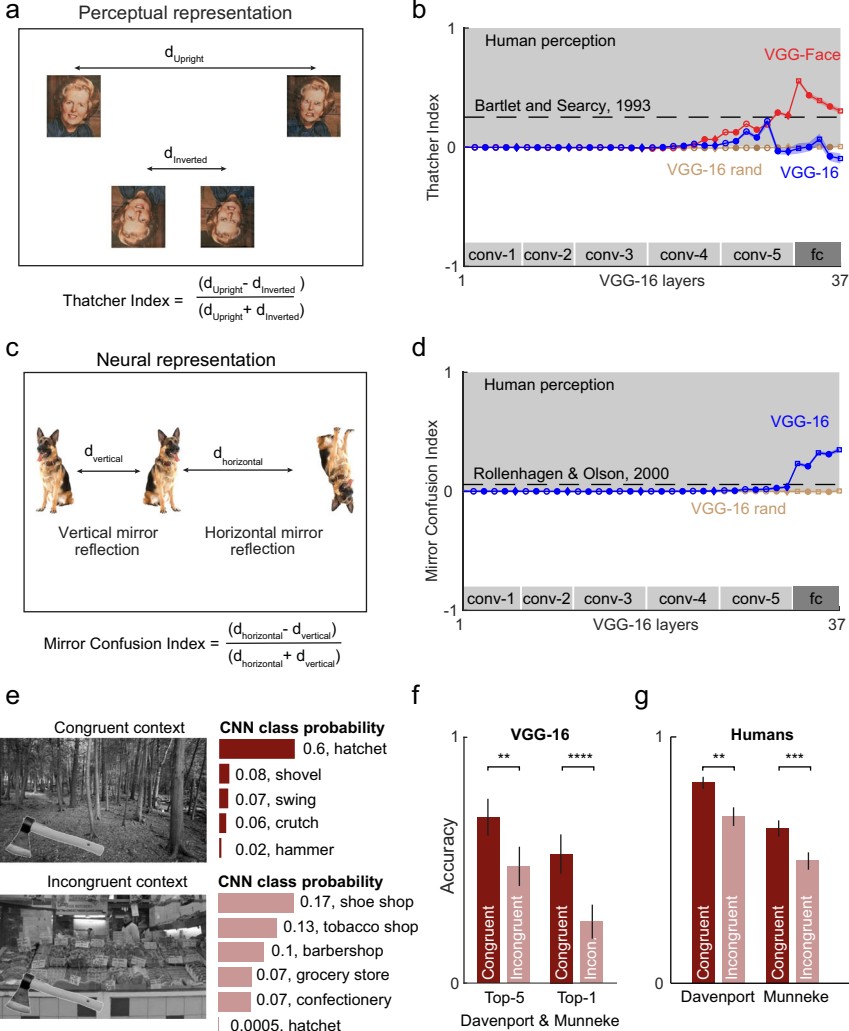

**Fig. 2 Object and scene regularities in deep networks. a** Perceptual representation of normal and Thatcherized faces in the upright and inverted orientations. **b** Thatcher index across layers of deep networks. For deep networks, we calculated the Thatcher index using Thatcherized faces from a recently published Indian face dataset (see the "Methods" section) across layers for the VGG-16 (*blue*), VGG-face (*red*), and a randomly initialized VGG-16 (*brown*). Error bars indicate s.e.m. across face pairs (*n* = 20). The *grey* region indicates human-like performance. Dashed lines represent the Thatcher index from a previous study[14], measured on a different set of faces (see the "Methods" section). **c** Neural representation of vertical and horizontal mirror images. Vertical mirror image pairs are closer than horizontal mirror image pairs. **d** Mirror Confusion Index (averaged across all objects) across layers for the pretrained VGG-16 network (*blue*) and a VGG-16 network with random weights (*brown*). Error bars indicates s.e.m. across stimuli (*n* = 50). Dashed lines represent the mirror confusion index estimated from monkey inferior temporal (IT) neurons in a previous study[18]. **e** An example object (*hatchet*) embedded in an incongruent context (*forest*) and in a congruent context (*supermarket*). The class probability returned by the VGG-16 network is shown beside each image for the top five guesses and for the correct object category (hatchet). **f** Accuracy of object classification by the VGG-16 network for congruent (*dark*) and incongruent (*light*) scenes, for top-5 accuracy (*left*) and top-1 accuracy (*right*). Error bars represent s.e.m. across all scene pairs (n = 40). Asterisks indicate statistical significance computed using the Binomial probability of obtaining a value smaller than the incongruent scene correct counts given a Binomial distribution with the congruent scene accuracy (** is *p* < 0.01, **** is *p* < 0.001). **g** Accuracy of object naming by humans for congruent (blue) and incongruent (red) scenes across two separate studies[19,20]. Error bars and statistical significance are taken from these studies (Davenport Set: ** indicates *p* < 0.01 for the main effect of congruence in a within-subject ANOVA on accuracy; Munneke Set: *** indicates *p* = 0.001 for accuracy in paired *t*-test).

scene (object + context) to the average feature vector for the object. By this measure, incongruent scenes were further away from the average compared to congruent scenes, and this effect arose only in later layers of all networks (Supplementary Section S4).

In sum, we conclude that deep networks also show scene incongruence like humans, but to a larger degree suggesting that they are more susceptible to contextual influences.

**Experiment 4: Do deep networks show multiple object normalization?** Next we asked whether individual units in deep networks conform to two general principles observed in single

neurons of high-level visual cortex. The first one is multiple object normalization, whereby the neural response to multiple objects is the average of the individual object responses at those locations[21]. This principle is illustrated in Fig. 3a. Note that this analysis is meaningful only for units that respond to all three locations: a unit in an early layer with a small receptive field would respond to objects at only one location regardless of how many other objects were present in the image. To identify units that are responsive to objects at each location, we calculated the variance of its activation across all objects presented at that location. We then performed this analysis on units that showed a non-zero response variance at all three

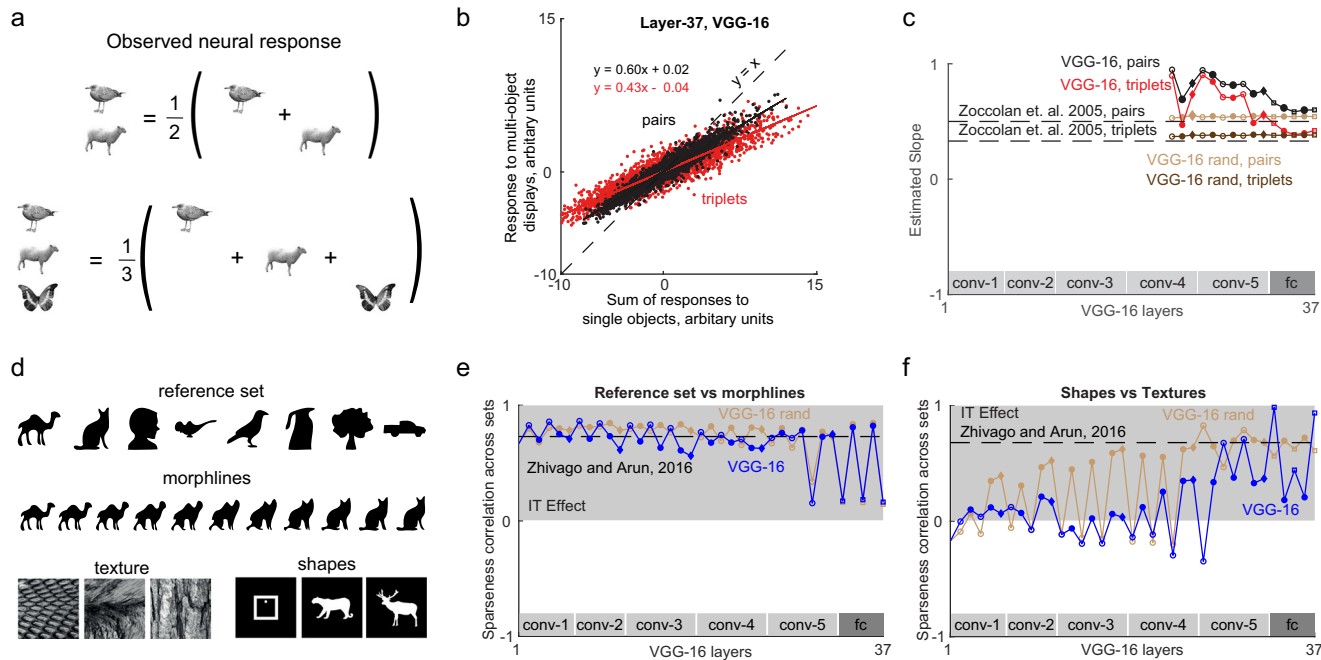

**Fig. 3 Single unit properties of deep networks. a** Schematic illustrating the general principle observed in neurons in high-level visual areas of the brain. The response of a neuron to multiple objects is typically the average of its responses to the individual objects at those locations. **b** Response to multiple-object displays plotted against the sum of the individual object responses for two-object displays (black) and three-object displays (red), across 10,000 units randomly selected from Layer 37 of the VGG-16 network. **c** Normalization slope plotted across layers for two object displays (*blue*) and three-object displays (*brown*) for the VGG-16 network and a randomly initialized VGG-16 (*brown*). The dashed lines depict the slopes observed in monkey IT neurons using a different stimulus set[21]. **d** Selected stimuli used to compare sparseness across multiple dimensions in a previous study of IT neurons[22]. **e** Correlation between sparseness on the reference set vs. along morphlines across units of each layer in the VGG-16 network (*blue*) and a VGG-16 with random weights (*brown*). The dashed line indicates the observed correlation in monkey IT neurons on the same set of stimuli. **f** Correlation between sparseness for textures and sparseness for shapes plotted across layers of the VGG-16 network (*blue*) and a VGG-16 with random weights (*brown*). The dashed line indicates the observed correlation in monkey IT neurons on the same set of stimuli.

locations, which meant units in Layer 23 (conv4.3) onwards for the VGG-16 network.

To assess whether deep networks show multiple object normalization, we plotted for each unit in a given layer its response to multiple objects against the sum of its responses to the individual objects. If there is multiple object normalization, the slope of the resulting plot should be 1/2 for two objects and 1/3 for three objects. The resulting plot is shown for Layer 37 of the VGG-16 network (Fig. 3b). The overall slope was 0.60 for two objects and 0.42 for three objects for all units. To evaluate this effect across layers, we plotted the two-object and three-object slopes obtained in this manner across layers (Fig. 3c). For the later layers we observed a nearly monotonic decrease in the slopes, approaching the levels observed in monkey high-level visual areas (Fig. 3c).

Interestingly, we observed perfect divisive-normalization in the randomly initialized VGG-16 (Fig. 3c). Upon closer investigation, we found that the activations for any two natural images were highly correlated (correlation for layer-37, mean ± s.d.: $r = 0.98 ± 0.01$), suggesting that these units were not selective to images. In other words, every image activates the network in the same way. As a result, the response to the combination AB and the individual images A and B separately would all be identical, giving rise to a perfect slope of 0.5 in the relationship between the response AB and the sum of responses A + B. Thus, the divisive normalization observed in the random network is a trivial outcome of its lack of image selectivity.

We conclude that deep networks exhibit multiple object normalization and image selectivity only after training.

**Experiment 5: Do deep networks show correlated sparseness?** In a recent study we showed that neurons in the monkey inferior temporal cortex have intrinsic constraints on their selectivity that manifests in two ways[22]. First, neurons that respond to fewer shapes have sharper tuning to parametric changes in these shapes. To assess whether units in the deep network VGG-16 show this pattern, we calculated the sparseness of each unit across a reference set of disparate shapes (Fig. 3d), and its sparseness for parametric changes between pairs of these stimuli (an example morph line is shown in Fig. 3d). This revealed a consistently high correlation between the sparseness on the reference set and the sparseness on the parametrically varying set, across units of each layer in the VGG-16 network (Fig. 3e). Second, we found that neurons that are sharply tuned across textures are also sharply tuned to shapes. To assess this effect across layers, we calculated the correlation across units between sparseness on textures with the sparseness on shapes. Although there was no such consistently positive correlation in the early layers, we did find a positive correlation in the later (conv5 and fc) layers (Fig. 3f). Importantly, we observed a similar, even stronger, trend in the randomly initialized VGG-16 (Fig. 3e, f). We observed similar results in other instances of VGG-16 (Supplementary Section S1) and in feedforward networks (Supplementary Section S2).

We conclude that deep networks show correlated sparseness along multiple dimensions just like neurons in high-level visual cortex, and that this is a property of their architecture, and not the training.

**Experiment 6: Do deep networks show Weber's law?** Next we asked whether deep networks are sensitive to relational properties in visual displays. The first and most widely known of

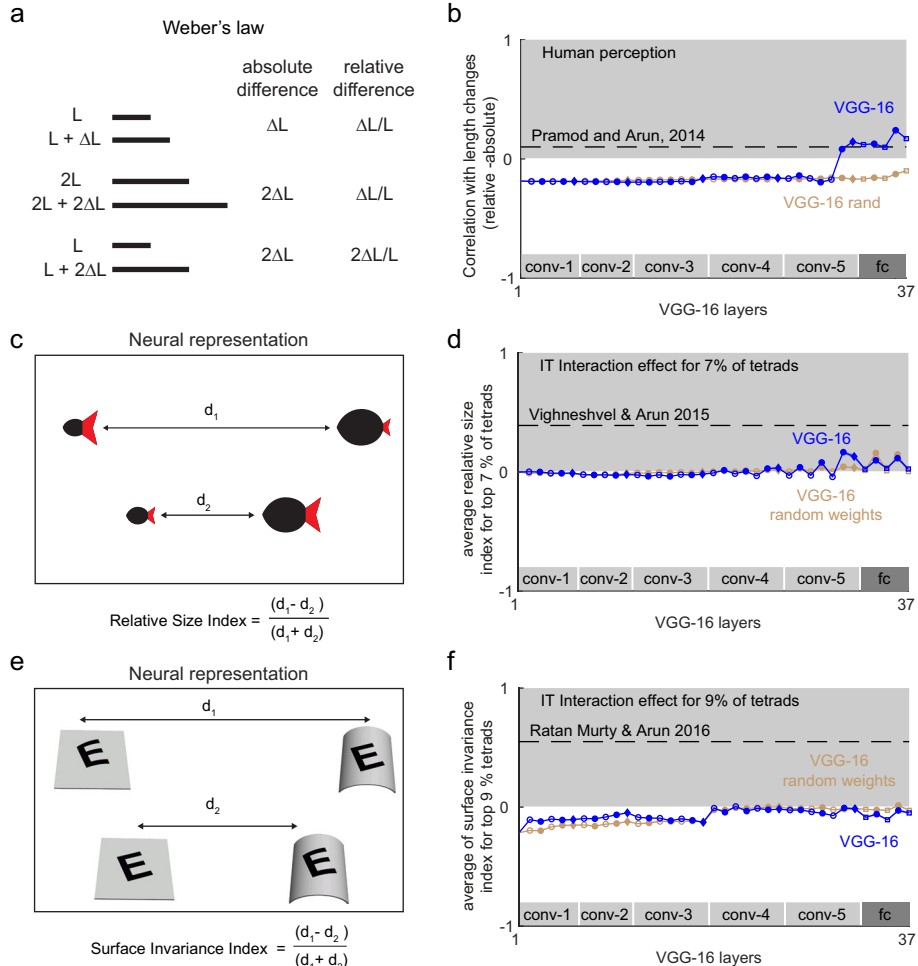

**Fig. 4 Relational properties in deep networks. a** Example illustrating the Weber's law for line length. Although the original statement of Weber's law is that the just-noticeable difference in length will depend on the baseline length, previous studies have shown that it also applies to perceptual distances[23]. In this formulation, the perceptual distance across pairs of lines differing in length will be more correlated with relative changes in length rather than absolute changes in length. **b** To calculate a single quantity that measures adherence to Weber's law, we calculated the correlation between distances and relative changes in length and subtracted the correlation between distances with absolute changes in length (see the "Methods" section). A positive difference indicates adherence to Weber's law (grey region). This difference in correlation is plotted across layers for line length in VGG-16 (*blue*) and a VGG-16 with random weights (brown). The dashed line indicates the value observed during human performing visual search on the same stimuli. **c** Schematic of the relative size encoding observed in monkey IT neurons[24]. Parts are coloured differently for illustration; in the actual experiments we used black silhouettes. For a fraction of neurons, the distance between two-part objects when both parts covary in size is smaller than the distance when they show inconsistent changes in size. Thus, these neurons are sensitive to the relative size of items in a display. **d** Relative size index across units with interaction effects (averaged across top 7% tetrads, error bars representing s.e.m.) across layers of the VGG-16 network (*blue*) and a VGG-16 with random weights (*brown*). The dashed line shows the strength of the relative size index estimated from monkey IT neurons on the same set of stimuli[24]. **e** Schematic of the surface invariance index observed in monkey IT neurons[25]. For a fraction of neurons, the distance between two stimuli with congruent changes in pattern and surface curvature is smaller than between two stimuli with incongruent pattern/surface changes. Thus, these neurons decouple pattern shape from surface shape. **f** Surface invariance index across units with interaction effects (averaged across top 9% pattern/surface tetrads, error bars representing s.e.m.) across layers of the VGG-16 network (*blue*) and a VGG-16 with random weights (*brown*). The dashed line depicts the surface invariance index estimated from monkey inferior temporal neurons on the same set of stimuli[25].

these is Weber's law, which states that sensitivity to just noticeable changes in any sensory quantity is proportional to the baseline level being used for comparison. The Weber's law for line length is depicted in Fig. 4a. This in turn predicts that the distance between any two lines differing in length should be proportional to the relative and not to the absolute change in length. In a previous study, we showed that this is true for humans in visual search for both length and intensity changes[23].

We therefore asked for the deep network VGG-16, whether pairwise distances between lines of varying length are correlated with absolute or relative changes in length (see the "Methods"

section). Specifically, if the correlation between pairwise distances and relative changes in length is larger than the correlation with absolute changes in length, we deemed that layer to exhibit Weber's law. This difference in correlation is positive for humans in visual search, and we plotted this difference across layers of the VGG-16 network (Fig. 4b). The correlation difference was initially negative in the early layers of the network, meaning that the early layers were more sensitive to absolute changes in length. To our surprise, however, distances in the later layers were sensitive to relative changes in length in accordance to Weber's law (Fig. 4b). Importantly, this trend was absent in the randomly initialized VGG-16 (Fig. 4b), suggesting that Weber's law arises

due to training and not solely due to the network architecture. We conclude that deep networks exhibit Weber's law for length, and that object classification training is sufficient to produce this effect.

**Experiment 7: Do deep networks encode relative size?** We have previously shown that neurons in high-level visual areas are sensitive to the relative size of items in a display[24]. Specifically, we found that, when two items in a display or two parts of an object undergo proportional changes in size, the neural response is more similar than expected given the two individual size changes. This pattern was present only in a small fraction (7%) of the neurons. This effect is illustrated in Fig. 4c. To explore whether this effect can be observed in a given layer of the deep network VGG-16, we performed a similar analysis. We selected units in a given layer with the strongest interaction between part sizes (see the "Methods" section) and calculated a relative size index of the form $(d1 - d2)/(d1 + d2)$, where $d1$ is the distance between stimuli with incongruent changes in size (when one part increases and the other decreases in size), and $d2$ is the distance between stimuli with congruent size changes (i.e. where both parts are scaled proportionally up or down in size). The relative size index was calculated for each tetrad (formed using images in which the size of each part was varied independently at two levels, resulting in a $2 \times 2$ tetrad) exactly as in the previous study on monkey IT neurons[24]. The relative size index for the VGG-16 network remained close to zero in the initial layers and increased modestly to a positive level in the later layers (Fig. 4d). However the size of this effect was far smaller than that observed in IT neurons, but nonetheless was in the same direction. Importantly, this trend was present albeit weakly in a randomly initialized VGG-16 (Fig. 4d), suggesting that it is partially a consequence of the network architecture and is strengthened by training. We conclude that deep networks represent relative size, and that this effect is present due to the network architecture and is strengthened by object recognition training.

**Experiment 8: Do deep networks decouple pattern shape from surface shape?** A recent study showed that IT neurons respond more similarly when a pattern and a surface undergo congruent changes in curvature or tilt[25]. This effect is illustrated for a pattern surface pair in Fig. 4e, where it can be seen that the distance between incongruent pattern–surface pairs (where the pattern and surface change in opposite directions) is larger than the distance between congruent pairs where the pattern and surface undergo similar changes. To assess whether the deep network VGG-16 shows this property, we identified units with increased interaction between surface curvature/tilt and pattern curvature/ tilt (see the "Methods" section) and calculated a surface invariance index of the form $(d1-d2)/(d1 + d2)$, where d1 is the distance between incongruent pairs (where the surface and pattern undergo changes in opposite directions), and $d2$ is the distance between congruent pairs (where the surface and pattern undergo similar changes). A positive value of this index for a given layer implies that the layer shows surface invariance. However, the surface invariance index was consistently below zero across layers for the VGG-16 network (Fig. 4f). We obtained similar trend even for the randomly initialized VGG-16 (Fig. 4f). We conclude that deep neural networks trained for object classification do not show surface invariance.

**Experiment 9: Do deep networks show 3d processing?** We are sensitive to three-dimensional shape and not simply two-dimensional contours in the image. This was demonstrated in an elegant experiment in which search for a target differing in 3D structure is easy whereas search for a target with the same difference in 2D features is hard[26,27]. This effect can be recast as a statement about distances in perceptual space as illustrated in Fig. 5a. All three pairs of shapes depicted in Fig. 5a differ in the same Y-shaped feature, but the two cuboids are more dissimilar because they differ also in 3D shape. To assess whether units in a given layer of the deep network show this effect, we calculated a 3d processing index of the form $(d1-d2)/(d1 + d2)$ where $d1$ is the distance between the cuboids and $d2$ is the distance between the two equivalent 2D conditions. A positive 3D processing index indicates an effect similar to human perception. However, we found that the 3D processing index was consistently near zero or negative across all layers of the VGG-16 network (Fig. 5b). We found similar results for the randomly initialized VGG-16 network (Fig. 5b). We conclude that deep networks are not sensitive to 3D shape unlike humans.

**Experiment 10: Do deep networks see occlusions as we do?** A classic finding in human perception is that we automatically process occlusion relations between objects[28]. Specifically, search for a target containing occluded objects among distractors that contain the same objects unoccluded is hard, whereas searching for the equivalent 2D feature difference is much easier (Fig. 5c, top row). Likewise, searching for a target that is different in the order of occlusion is hard, whereas searching for the equivalent 2D feature difference is easy (Fig. 5c, bottom row). These observations demonstrate that our visual system has a similar representation for occluded and unoccluded displays.

We therefore asked whether similar effects are present in the VGG-16 network, by calculating an occlusion index of the form $(d2 - d1)/(d2 + d1)$ where d1 is the distance between two displays that are equivalent except for occlusion, and $d2$ is the distance between equivalent displays with the same 2D feature difference. A positive occlusion index implies an effect similar to human perception. For humans, we calculated the occlusion index using the reciprocal of search slopes reported in previous studies[28], since the reciprocal of search times have been shown to behave like mathematical distance metric[29]. For the VGG-16 network, the occlusion index remained consistently negative but approached zero across layers, for both the occlusion and depth ordering effects (Fig. 5d). By contrast, we observed no such increasing trend in the randomly initialized VGG-16 network (Fig. 5d). Thus, object classification training modifies these effects but still does not make them human-like. We conclude that deep networks do not represent occlusions and depth ordering the way we do.

**Experiment 11: Do deep networks see object parts the way we do?** We not only recognize objects but are able to easily describe their parts. We conducted two related experiments to investigate part processing in deep networks. In Experiment 11A, we compared deep network feature representations for whole objects and for the same object with either natural or unnatural part cuts. In perception, searching for an object broken into its natural parts with the original object as distractors is much harder than searching for the same object broken at an unnatural location[30]. This result is depicted schematically in terms of the underlying distances (Fig. 6a). To assess whether the VGG-16 network also shows this part decomposition, we calculated a part processing index of the form $(d_u-d_n)/(d_u + d_n)$ where $d_u$ is the distance between the original object and the object broken at an unnatural location, and $d_n$ is the distance between the original object and the same object broken at a natural location. A positive part processing index implies an effect similar to that seen in

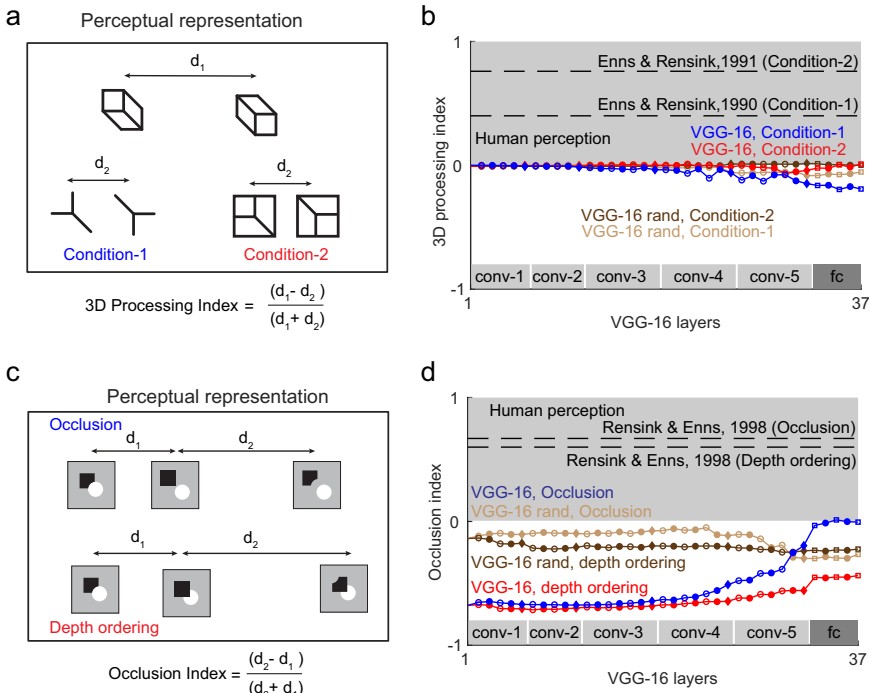

**Fig. 5 3D processing in deep networks. a** Perceptual representation of sensitivity to 3D shape[27]. Three equivalent image pairs are shown in perceptual space. The first image pair (with distance marked $d1$) consists of two cuboids at different orientations, and represents an easy visual search, i.e. the two objects are dissimilar. The second pair (marked with distance $d2$) contains the same feature difference as in the first pair, but represents a hard search, i.e. is perceived as more similar. Likewise, the third pair (also marked with distance $d2$), with the same feature difference as the first image pair, is a hard search, i.e. perceived as similar. **b** 3D processing index for the VGG-16 network across layers, for condition 1 (*blue*) and condition 2 (*red*), and for the VGG-16 with random weights (dark brown and light brown). Dashed lines represent the estimated human effect measured using visual search on the same stimuli. **c** Perceptual representation of occluded shapes. *Top:* A square alongside a disk is perceptually similar to a display with the same objects but occluded, but dissimilar to a 2D control image with an equivalent feature difference. *Bottom:* A square occluding a disk or disk occluding square are perceptually similar, but dissimilar to an equivalent 2d control with the same set of feature differences. **d** Occlusion index for the occlusion (*blue*) and depth ordering (*red*) effects for each layer of the VGG-16 network, and for the randomly initialized VGG-16 (*dark brown* and *light brown*). Dashed lines represent the effect size in humans measured using visual search on the same stimuli.

perception. In perception, we calculated the part processing index using the reciprocal of search slopes observed in humans during visual search. The part processing index across layers of the VGG-16 network is depicted in Fig. 6b. We found that the index becomes positive in the intermediate layers, but becomes negative in the subsequent layers (conv4/conv5 onwards). By contrast, we observed no such decreasing trend in the randomly initialized VGG-16 network (Fig. 6b), suggesting that object classification training causes the part processing index to become negative.

In Experiment 11B, we asked what happens to objects that can be decomposed into two possible ways without introducing a break (Fig. 6c). In visual search, search between pairs of whole objects is explained better using its natural parts than its unnatural parts[31]. In other words, models that explain whole-object dissimilarities using the constituent parts performed better when the parts were the natural parts compared to the unnatural parts.

To capture this effect, we defined the natural part advantage as the difference in model correlation (see the "Methods" section) between natural and unnatural parts for the distances calculated in any given layer of the deep networks. A positive value indicates an effect similar to perception. This natural part advantage is shown across layers of the VGG-16 network in Fig. 6d. It can be seen that there is little or no advantage for natural parts in most layers except temporarily in the later layers (conv5/fc). We observed similar trends for the randomly initialized VGG-16 network (Fig. 6d).

Based on Experiments 11A and 11B, we conclude that the VGG-16 network shows no systematic part decomposition with or without training.

**Experiment 12: Do deep networks show a global shape advantage?** In perception a classic finding is that we see the forest before the trees, i.e. we can detect global shape before local shape[32,33]. We can recast this effect into a statement about distances in perception: the distance between two hierarchical stimuli differing only in global shape will be larger than the distance between two such stimuli differing only in local shape. This is depicted schematically in Fig. 6e. To calculate a single measure for this effect, we defined a global advantage index as $(d_{global} - d_{local})/(d_{global} + d_{local})$, where $d_{global}$ is the average distance between all image pairs differing only in global shape and $d_{local}$ is the average distance between all image pairs differing only in local shape. A positive global advantage index implies an effect similar to perception. For perception we measured the global advantage by taking the reciprocal of search times as the perceptual distance[29].

The global advantage index is depicted across layers of the VGG-16 network in Fig. 6f. While there is a slight global advantage in the initial layers, the network representation swings rapidly in the later layers towards the opposite extreme, which is a local advantage. This is likely due to increased pooling in the higher layers, but interestingly the pre-trained VGG-16 network shows the opposite pattern. Thus, it appears that object classification training abolishes

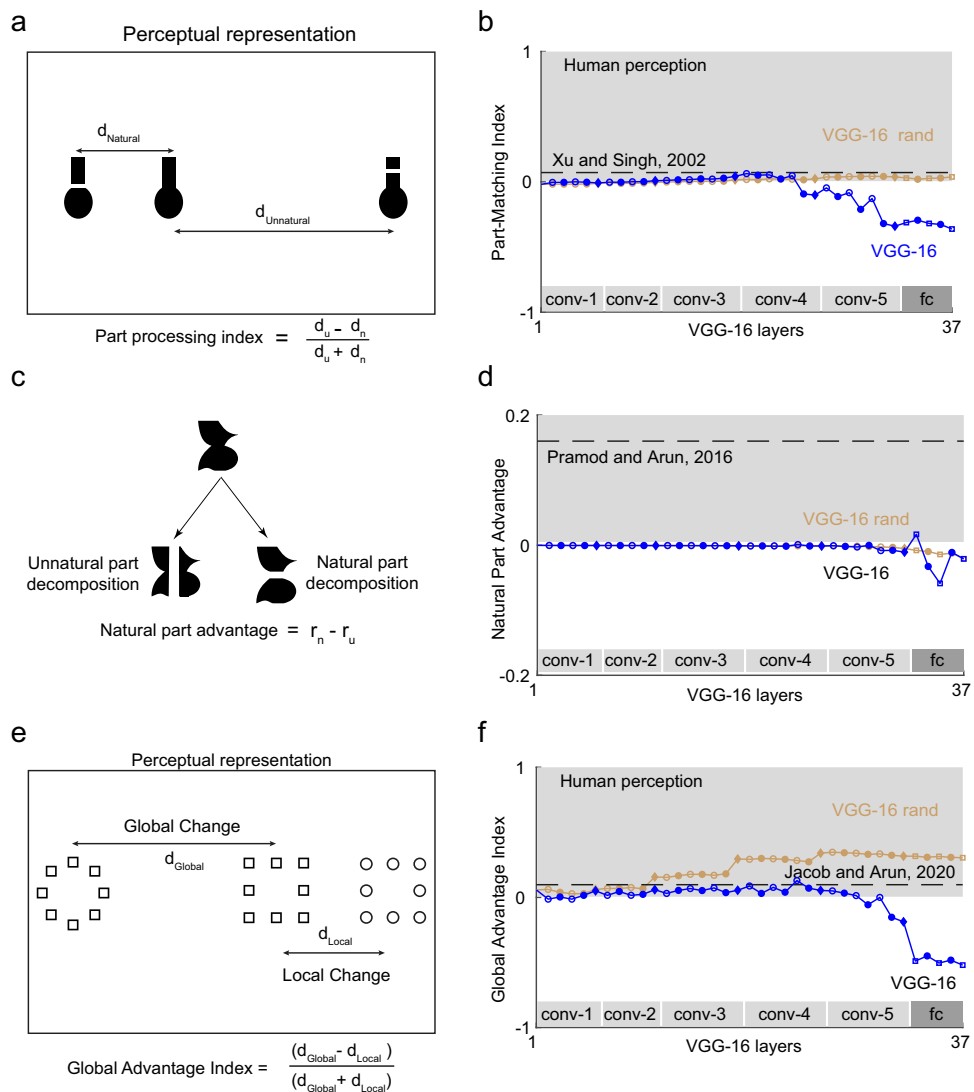

**Fig. 6 Part-whole relations in deep networks. a** Schematic showing the perceptual representation of objects with a break introduced either at natural or unnatural part cuts. **b** Part processing index across layers of the VGG-16 network (*blue*) and VGG-16 with random weights (*brown*). The dashed line represents the effect size estimated from human visual search on the same stimuli[30]. **c** Schematic showing how the same object can be broken into either natural or unnatural parts. The natural part advantage is calculated as the difference in correlation between part-sum models trained to predict whole-object dissimilarities using the parts (see he "Methods" section). **d** Natural part advantage across layers of the VGG-16 network (*blue*) and VGG-16 with random weights (*brown*). The dashed line represents the effect size estimated from human visual search on the same stimuli[31]. **e** Perceptual representation of hierarchical stimuli. The left and middle images differ only in global shape whereas the middle and right images differ only at the local level. According to the global advantage effect, a change in global shape is more salient than a change in local shape. **f** Global advantage index across layers of the VGG-16 network (*blue*) and VGG-16 with random weights (*brown*). The dashed line represents the effect size estimated from human visual search on the same stimuli[63].

the global advantage that is intrinsic to the network architecture. We speculate that this local advantage might arise because of the demands of distinguishing between highly similar categories present in the ImageNet dataset (e.g. there are 90 categories of dogs among the total of 1000 categories in ImageNet). Testing this possibility will require training the VGG-16 architecture on highly distinctive object classes.

Interestingly, the randomly initialized VGG-16 network showed a clear global advantage (Fig. 6f). We conclude that training deep networks for object classification abolishes the global advantage and introduces a local advantage.

## Discussion
Here we report qualitative similarities and differences in object representations between brains and deep neural networks trained

for object recognition. We find that some properties are present even in randomly initialized networks. Many others were present in feedforward deep neural networks after training on object classification. Yet others were absent even after training. These results are summarized in Table 1. Our findings generalized across other instances of VGG-16 (Supplementary Section S1), to other feedforward neural networks (Supplementary Section S2) and across distance metrics (Supplementary Section S3). They however apply only to feedforward neural networks trained for object classification.

These findings are important for several reasons. First, they describe qualitative similarities and differences in object representations between brains and deep networks trained for object recognition. These findings clarify the conditions under which deep networks can and cannot be considered accurate models of

**Table 1 Presence/absence of each effect across deep networks tested.**

| Perceptual effect | VGG-16 random | VGG-16 | AlexNet | GoogLeNet | ResNet-50 | ResNet-152 |
|---|---|---|---|---|---|---|
| Thatcher effect | No | No | No | Yes | No | No |
| Mirror confusion | No | Yes | Yes | Yes | Yes | Yes |
| Scene incongruence | No | Yes | Yes | Yes | Yes | Yes |
| Multiple objects | No | Yes | Yes | Yes | Yes | Yes |
| Correlated sparseness | Yes | Yes | Yes | Yes | Yes | Yes |
| Weber's law | No | Yes | Yes | Yes | Yes | Yes |
| Relative size | Yes | Yes | Yes | Yes | Yes | Yes |
| Surface invariance | No | No | No | No | No | No |
| 3D processing | No | No | No | No | No | No |
| Occlusion | No | No | No | No | No | No |
| Depth ordering | No | No | No | No | No | No |
| Object parts | No | No | No | No | No | No |
| Global advantage | Yes | No | No | No | No | No |

biological vision. Second, they show that some effects arise in deep networks solely due to their architecture, some arise after object classification training, and others are absent even despite training. Third, the missing properties could be incorporated as training or architecture constraints on deep networks to yield better or more robust performance. Below we discuss our findings in the context of the existing literature.

We first address some general concerns. First and foremost, it could be argued that our results are based on testing with artificial objects or images, and that it is unreasonable to expect deep networks to respond sensibly to unnatural images. However, these concerns apply equally to humans as well, who in fact do respond sensibly to these artificial displays often without any prior exposure. Indeed, there is a long tradition in psychology and neuroscience of using artificial images to elucidate visual processing[34]. Second, it could be argued that deep networks could potentially be trained to report all the tested properties. However, such a finding would only be circular if the network did indeed exhibit the property it was trained for. We do note however that it would be interesting if deep networks were unable to learn certain properties. Indeed, certain relational properties have been reported as difficult to learn by computer vision algorithms[12], although this study did not evaluate deep neural networks. By contrast, our results are more insightful, since they reveal which emergent properties arise in deep networks without explicit task training.

**Properties present in randomly initialized networks**. We have found that several properties are present in randomly initialized deep networks, such as correlated sparseness, relative size encoding, and global advantage. It suggests that the network architecture itself can give rise to interesting properties. It is consistent with the important but neglected finding that even a randomly initialized network can generate useful features[35,36]. Even early visual responses in human MEG data were recently predicted using a randomly initialized deep neural network[37]. We speculate that additional object properties might be incorporated by modifying the architecture of deep networks.

**Properties present in deep networks trained for object recognition**. We found that many properties were present in deep networks only after training on object classification. Our finding that deep networks exhibit Weber's law is puzzling at first glance. Why would the demands of recognizing objects require sensitivity to relative changes? One possibility is that object recognition requires a representation invariant to changes in size, position, viewpoint and even illumination of objects in the image, which in turn requires processing all object features relative to the surrounding features. This could be tested by training deep networks

on controlled sets of images with variations of one kind but not the other. It is also possible that there are other task requirements that could give rise to Weber's law[38]. Our finding that deep networks exhibit the Thatcher effect, mirror confusion and scene incongruence are consistent with them being sensitive to image regularities in scenes. In fact, deep networks may be over-reliant on scene context, because it showed a larger drop in accuracy for incongruent scenes compared to humans (Fig. 2f). This is consistent with a previous study in which human scene expectations benefited deep network performance[39].

**Properties absent in deep networks**. We have found that both randomly initialized and object-trained deep networks did not exhibit 3D processing, occlusions and surface invariance, suggesting that these properties might emerge only with additional task demands such as evaluating 3D shape. Likewise, the absence of any part processing or global advantage in deep networks suggests that these too might emerge with additional task demands, such as part recognition or global form recognition[40]. We have found that deep networks do not show a global advantage effect but instead seem to process local features. This finding is surprising considering that units in later layers receive convergent inputs from the entire image. However, our finding is consistent with reports of a bias towards local object texture in deep networks[41]. It is also consistent with the large perturbations in classification observed when new objects are added to a scene[42], which presumably change the distribution of local features. Our finding that deep networks experience large scene incongruence effects is therefore likely to be due to mismatched local features rather than global features. Indeed, incorporating scene expectations from humans (presumably driven by global features) can lead to substantial improvements in object recognition[39]. Finally, a reliance on processing local features is probably what makes deep networks detect incongruously large objects in scenes better than humans[43]. We speculate that training on global shape could make deep networks more robust and human-like in their performance.

Finally, we note that some properties temporarily emerged in intermediate layers but are eventually absent in the final fully connected layers important for classification (Figs. 2b, 4d, 6f). We do not have an adequate explanation, but it is possible that these reflect intermediate computations required to create the final representation needed for classification. In the brain likewise, there could be many properties in the intermediate visual areas like V4 that are not necessarily present in early or final stages[44]. Indeed the early, intermediate, and later stages of ventral pathway in the primate brain coarsely correspond to early, middle, and later layers of a deep neural network[10,15]. Elucidating these

intermediate representations is therefore an interesting topic for further study.

**How can deep networks be improved using these insights?** Can learning on different datasets and tasks influence the features learned in a deep neural network and bring them closer to the representations found in the brain? Recent evidence shows that deep neural networks trained on specific tasks (like scene parsing) can explain the responses in functionally specific brain regions (like the occipital place area that is known to be involved in recognizing navigational affordances) better than a deep neural network trained on a different task[45]. In addition, advances in unsupervised learning have led to deep neural networks with feature representations that can not only transfer better to other tasks but also predict neural data better than supervised models[46,47]. Yet other studies have attempted to improve deep networks by augmenting them with perceptual or neural representations[39,48]. How can deep networks be made to match neural and perceptual representations? There could be several ways of doing so. The first and perhaps most promising direction would be to explicitly train deep networks to produce such properties in addition to categorization[49]. Another alternative would be to train deep networks on tasks such as navigation or agent–object interaction rather than (or in addition to) object recognition as this is ostensibly what humans also do[50,51].

Finally, we note that deep networks are notoriously susceptible to adversarial attacks. State-of-the-art deep neural networks have been shown to fail catastrophically when input images are subjected to carefully constructed changes that are barely perceivable to the human eye[52,53]. Likewise, deep networks can give erroneous predictions on completely nonsensical images[54] and produce natural image metamers that are unrecognizable for humans[55]. What could underlie such unusual behaviour? One possible reason could be the tendency for deep networks to prioritize local features as described earlier. Another is that these adversarial images can contain some weakly relevant features used by humans[56] and could be adversarial even for humans at brief viewing durations[57]. We speculate that training deep networks to exhibit all the perceptual and neural properties described in this study might not only improve their performance but also make them more robust to adversarial attacks.

## Methods

**VGG-16 network architecture and training**. All experiments reported in the main text were performed on the VGG-16 network, a feedforward pre-trained deep convolutional neural network trained for object classification on the ImageNet dataset[58]. In Supplementary Sections 1 and 2, we show that these results generalize to other instances of VGG-16 as well as to other feedforward architectures. We briefly describe the network architecture here, and the readers can refer to the original paper for more details[16]. The network takes as input an RGB image of size $224 \times 224 \times 3$, and returns a vector of confidence scores across 1000 categories. We subtracted the mean RGB value across all images (mean values across all images: $R = 123.68$, $G = 116.78$, $B = 103.94$). In the network, the image is passed through a stack of convolutional filters (Fig. 1c), where the initial layers have small receptive field ($3 \times 3$) and later layers are fully connected. A non-linear rectification (ReLu) operation is performed after each convolution operation. Five max-pooling layers are present to spatial pool the maximum signal over a $2 \times 2$ window of neurons. We used the MATLAB-based MatConvNet software platform[59] to extract features and do the analysis. In addition to VGG-16, we also used VGG-face which has the same architecture but trained instead on face identification[17].

**Feature extraction**. We passed each image as input to the network, and stored the activations of each layer as a column vector. Hence, a single image we will have 37 feature vectors (one column vector from each layer). To calculate the distance between images A and B, we calculated the Euclidean distance between the corresponding activation vectors.

For each experiment, we defined a specific measure and plotted it across layers with a specific chain of symbols as shown in Fig. 1c. Symbols used indicate the underlying mathematical operations performed in that layer: unfilled circle for convolution, filled circle for ReLu, diamond for maxpooling and unfilled square for

fully connected layers. Broadly, filled symbols denote linear operations and unfilled ones indicate non-linear operations.

**Experiment 1: Thatcher effect**. The stimuli comprised 20 co-registered Indian faces (19 male, 1 female) from the IISc Indian Face Dataset[5]. All faces were grayscale, upright and front-facing. To Thatcherize a face, we inverted the eyes and mouth while keeping rest of the face intact. We implemented inversion by first registering facial landmarks on frontal faces using an Active appearance model-based algorithm[60]. Briefly, this method models face appearance as a two-dimensional mesh with 76 nodes, each node represents local visual properties of stereotyped locations such as corners of eyes, nose, and mouth. We then defined bounding boxes for left and right eye as well as mouth, by identifying landmarks that correspond to the four corners of each box. We then locally inverted eye and mouth shape by replacing the top row of eye or mouth image pixels by the last row and likewise repeating this procedure for each pair of equidistant pixels rows above and below the middle of the local region. The inversion procedure was implemented as a custom script written in MATLAB.

To calculate a single measure for the Thatcher effect, we calculated the Thatcher index defined as $\frac{d_{upright} - d_{inverted}}{d_{upright} + d_{inverted}}$, where $d_{upright}$ is the distance between an normal face and Thatcherized face in upright orientation and $d_{inverted}$ is the distance between a normal face and Thatcherized face in inverted orientation. We estimated the Thatcher index for humans from the similarity ratings reported from humans albeit on a different set of faces[14]. We calculated the Thatcher index after converting the similarity rating (humans gave a rating between 1 and 7 on pair of images) into a dissimilarity rating (dissimilarity rating = 7−similarity rating).

**Experiment 2: Mirror confusion**. The stimuli consisted of 100 objects (50 naturalistic objects and 50 versions of these objects made by rotating each one by 90°). We created a horizontal and vertical mirror image of each object. Keeping both original and 90° rotated versions of each object ensured that there was no spurious difference between vertical vs horizontal images simply because of objects being horizontally or vertically elongated. We then gave as input the original image and the two mirror images to the VGG-16 network and calculated for each layer the distance between the object and two mirror images.

To calculate a single measure for mirror confusion, we defined a mirror confusion index of the form $\frac{d_{horizontal} - d_{vertical}}{d_{horizontal} + d_{vertical}}$, where $d_{horizontal}$ is the distance between an object and its horizontal mirror image and $d_{vertical}$ is the distance between an object and its vertical mirror image. We estimated the strength of mirror confusion index in the brain using previously published data from monkey IT neurons[18]. Specifically, we took $d_{horizontal}$ to be the reported average firing rate difference between the original objects and its horizontal mirror image, and analogously for $d_{vertical}$.

**Experiment 3: Scene incongruence**. The stimuli consisted of 40 objects which was taken from previous studies: 17 objects were from the Davenport study[19] and the remaining 23 from the Munneke study[20]. We discarded 11 objects from Davenport study and 5 objects from Munneke study since they did not have a matching category label in the ImageNet database. Each object was embedded against a congruent and an incongruent background.

We measured the classification accuracy (Top-1 and Top-5) of the VGG-16 network for the objects pasted onto congruent and incongruent scenes. The final layer of VGG-16 (Layer 38) returns a probability score for all 1000 categories in the ImageNet database. The top-1 accuracy is calculated as the average accuracy with which the object class with the highest probability matches the ground-truth object label. The top-5 accuracy is calculated as the average accuracy with which the ground-truth object is present among the object classes with the top 5 probability values. We report the human (object naming) accuracy on the same dataset from previous studies[19,20].

**Experiment 4: Multiple object normalization**. The stimuli consisted of 49 natural grayscale objects. To investigate responses to individual objects and to multiple objects, we created 147 singleton images by placing each of the 49 objects at 3 possible locations (Fig. 4a). We then created 200 object pairs and 200 object triples by random selection. We extracted features for all images (singletons, pairs and triplets) from every layer of the CNN. We selected a unit for further analysis only if the unit responded differently to at least one of the images in all the three positions. We then plotted the sum of activations of selected units in a layer to the singleton images against the activation for the corresponding pairs (or triplets). The slope of this scatterplot across layers was used to infer the nature of normalization in CNNs —a slope of 0.5 for pairs and 0.33 for triplets indicated divisive normalization matching that observed in high-level visual cortex.

**Experiment 5: Selectivity along multiple dimensions**. Here we used the stimuli used in a previous study to assess the selectivity of IT neurons along multiple dimensions[22]. These stimuli consisted of 8 reference shapes (Fig. 3d; top row) and created intermediate parametric morphs between pairs of these shapes (Fig. 3d; example morph between camel to cat). In addition, to compare texture and shape selectivity, we used 128 natural textures and 128 silhouette shapes.

As before we calculated the activation of every layer of the VGG-16 network to each of the above stimuli are input. Visually active neurons were selected by finding units with a non-zero variance across this stimulus set. We found the visually active neurons for each set separately and we selected a unit for further analysis only if that unit is visually active for both sets. The response of each unit was normalized between 0 and 1. We then calculated the sparseness of each unit across different stimulus sets: the reference set, the four morphlines, shape set and texture set. For a given stimulus set with responses $r_1, r_2, r_3, ..., r_n$, where $n$ is the number of stimuli,

the sparseness is defined as follows: $S = \left(1 - \frac{\left(\frac{\sum r_i}{n}\right)^2}{\sum \frac{r_i^2}{n}}\right) / \left(1 - \frac{1}{n}\right)$ [22,61]. We then

calculated the correlation across neurons between the sparseness on one stimulus set versus another stimulus set.

**Experiment 6: Weber's Law.** To test for the presence of Weber's law in the deep network, we created images of a rectangular bar varying in the length. We selected bar lengths such that the length difference computed on pairs of images spanned a wide range both in terms of absolute as well as relative differences.

For each layer of the neural network, we extracted the activations for each image and then computed the pairwise dissimilarity for all image pairs. We then computed the correlation $r_{abs}$ between pairwise dissimilarities and absolute length differences (i.e. between $d_{ij}$ and $\Delta_{ij} = |L_i - L_j|$ across all images $i$ & $j$, where $d_{ij}$ is the distance between and $L_i, L_j$ are the bar lengths. We also calculated the correlation $r_{rel}$ between pairwise dissimilarity and relative length differences (i.e. between $d_{ij}$ and $\Delta_{ij} = \frac{|L_i - L_j|}{0.5*(L_i + L_j)}$ across all images $i$ & $j$). The difference $r_{rel} - r_{abs}$ is expected to be positive if the representation follows Weber's law. For humans, we calculated this correlations by taking the distance to be the reciprocal of search time for the corresponding pairs of lines.

We also analysed deep networks for the presence of Weber's law for image intensity, but found highly inconsistent and variable effects. Specifically, the pre-trained VGG-16 network showed Weber's law for low image intensity levels but not for high intensity levels.

**Experiment 7: Relative size.** We used the stimuli used in a previous study to test whether units in the VGG-16 network encode relative size[24]. This stimulus set consisted of 24 tetrads. A sample tetrad is shown in Fig. 2d, with the stimuli arranged such that images that elicit similar activity are closer.

In our previous study[24], only a small fraction of neurons (around 7%) encoded relative size. To identify similar neurons in the deep network, we first identified the visually responsive units by taking all units with a non-zero variance across the stimuli. For each unit and each tetrad, we calculated a measure of size interactions of the form abs($r11 + r22 - r12 - r21$), where $r11$ is the response to both parts at size 1, $r12$ is the response to part 1 at size 1 and part 2 at size 2, etc. We then selected the top 7% of all tetrads with the largest interaction effect. Note that this step of selection does not guarantee the direction of the relative size effect. For the selected tetrads, we calculated the relative size index, defined as $\frac{d_1 - d_2}{d_1 + d_2}$ where $d_1$ and $d_2$ are distances between the incongruent and congruent stimuli respectively.

**Experiment 8: Surface invariance.** The stimuli consisted of six patterns super-imposed on four surfaces. Each pattern–surface pair was used to create a tetrad of stimuli as depicted in Fig. 4e. The full stimulus set consisted of 24 tetrads, which were a subset of those tested in our previous study[25].

In each VGG-16 layer, we selected visually responsive neurons and normalized their responses across all stimuli as before in Experiment 5. We then selected the top 9% of all tetrads with an interaction effect calculated as before, as with the previous study[25]. For all the selected tetrads we calculated the surface invariance index, defined as $\frac{d_1 - d_2}{d_1 + d_2}$ where $d_1$ and $d_2$ are distances between incongruent and congruent stimuli.

**Experiment 9: 3D processing.** We investigated 3D processing in the VGG-16 network by comparing line drawing stimuli used in a previous perceptual study[27]. We compared three pairs of shapes: cuboid, cube and frustum of square in iso-metric view with the corresponding Y junctions. For each shape, we calculated three distances between equivalent shape pairs with the same feature difference (Fig. 5a). We calculated a 3D processing index, defined as $\frac{d_1 - d_2}{d_1 + d_2}$ where $d_1$ and $d_2$ are distances between the 3D shape and control conditions, respectively. To calculate the index for humans, we took the distances as the reciprocal of search slopes reported in the study for each condition.

**Experiment 10: Occlusions.** We recreated the stimulus set used in a previous study[62] as depicted in Fig. 5c. As before we compared the distance between two pairs of shapes: a pair that differed in occlusion status (occluded vs. unoccluded, or two images that differed in their order of occlusion), and an equivalent 2D feature control containing the same feature difference. We then calculated an occlusion index defined as $\frac{d_1 - d_2}{d_1 + d_2}$ where $d_1$ and $d_2$ are the distances between the occluded and

control conditions, respectively. To calculate the index for humans, we took the distances as the reciprocal of search slopes reported in the study for each condition.

**Experiment 11: Object parts.** We performed two experiments to investigate part processing in deep networks. In Experiment 11A, we tested what happens when a break is introduced into an object at a natural cut or an unnatural cut[30]. The critical comparison is shown in Fig. 6a. For each layer of the CNN, we extracted features for the three objects and computed the distance of the intact object with each of the broken objects ($d_n$ and $d_u$ denote distances to the broken objects with natural and unnatural parts, respectively). We then computed a part processing index, defined as $\frac{d_u - d_n}{d_u + d_n}$.

In Experiment 11B, we asked whether whole object dissimilarities computed on CNN feature representations could be understood as a linear combination of dissimilarities between their natural or unnatural part decompositions as reported previously for visual search[31]. We considered seven whole objects that could be broken down into either natural or unnatural parts and recombined the parts to form other objects. That is, we created two sets each containing 49 objects made either from natural or unnatural parts of the original seven objects. We then selected 492 pairs of objects from each set (including all 21 pairs from the common set) and calculated the feature distances from each layer of the CNN. We fit a part summation model to explain pairwise whole-object distances as a function of pairwise part relations[31].

Briefly the part summation model uses pairwise relations between parts to predict pairwise dissimilarities between objects. For two objects AB and CD (made of parts A, B, C, D), the dissimilarity is given by $d(AB,CD) = C_{AC} + C_{BD} + X_{AD} + X_{BC} + W_{AB} + W_{CD} + constant$, where $C_{AC}$ & $C_{BD}$ represent part relations between corresponding parts of the two objects, $X_{AD}$ & $X_{BC}$ represent part relations between parts at opposite locations in the two objects and $W_{AB}$ & $W_{CD}$ represent part relations within each object. Since there are 49 objects, and therefore $^{49}C_2 = 1176$ pairwise dissimilarities between objects, there are 1176 equations that can be written for the part-sum model. However, since there are only 7 parts, each set of terms (C, W & X terms) have only 7C2 = 21 part relations. This linear system of equations thus contains 1176 equations and $21 \times 3 + 1 = 64$ unknown terms that can be estimated using linear regression.

Having fit the part-sum model to each layer, we compared model performance on the 21 pairwise distances between the common objects. We denoted by $r_{natural}$ the correlation between observed and predicted distances assuming natural part decomposition and by $r_{unnatural}$ the model correlation assuming unnatural part decomposition. The natural part advantage was computed as ($r_{natural} - r_{unnatural}$). The same measure was computed for human perception.

**Experiment 12: Global shape advantage.** We created a set of 49 hierarchical stimuli by combining seven shapes at global scale and the same seven shapes at the local scale[63]. For human perception, we calculated distances as the reci-procal of the average search time for each image pair. For CNNs, we extracted features from each layers and calculated the Euclidean distance between all pairs of stimuli. We calculated the global distance as the mean distance between image pairs having only global change. Similarly, we calculated the local distance as the mean distance between image pairs having only local change. A sample global/local change pair is shown in Fig. 6e. We calculated a global advantage index as $\frac{d_{Global} - d_{Local}}{d_{Global} + d_{Local}}$.

**Reporting summary.** Further information on research design is available in the Nature Research Reporting Summary linked to this article.

## Data availability
All data used to produce the results are publicly available at https://osf.io/35fmh/.

## Code availability
All codes used to produce the results are publicly available at https://osf.io/35fmh/.

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

## Acknowledgements

This research was supported by the DBT/Wellcome Trust India Alliance Senior Fellowship awarded to S.P.A. (Grant# IA/S/17/1/503081), and by travel grants to G.J. from the Department of Science and Technology, Government of India and from the Pratiksha Trust.

## Author contributions

G.J., R.T.P., H.K., and S.P.A. designed the study, G.J. performed Experiments 5,7,8–12 and replicated the analyses for all experiments, H.K. and G.J. performed Experiments 1–3, R.T.P. performed Experiments 4, 6 & 11, and G.J. & S.P.A. wrote the manuscript with inputs from R.T.P. & H.K.

## Competing interests

The authors declare no competing interests.
