## [Peer Review File · Nature Communications]

Reviewers' Comments:

Reviewer #1:

Remarks to the Author:

Review of Jacob et al.

I apologize to the authors and editor for taking a long time to review this paper.

This paper evaluates similarities between biological visual systems and artificial neural networks trained to recognize objects. Artificial networks have recently become favored models for ventral visual stream processing, based on their ability to predict neural responses and replicate patterns of behavioral errors. This paper provides further tests of these similarities by assessing twelve different phenomena documented in the behavior or neural responses of primate visual systems. With a few exceptions, the general methodology is to measure distances between the activations evoked by pairs of images that are perceived as more or less similar by humans. The findings are that some of the twelve phenomena are evident in the artificial neural networks while others are not. The authors argue that these reveal some qualitative differences with biological vision.

Overall assessment:

This paper has some interesting results. There is a lot of interest right now in comparisons between artificial neural networks and biological sensory systems, and these results will find an audience because of that. The weaknesses of the paper are that 1) some of the phenomena seem more interesting/important than others, 2) the phenomena are not obviously chosen in a principled way, and as a result the comparisons seem fairly disconnected from any sort of theory, 3) the human and model measurements are not made with a consistent methodology, and 4) many important control results are buried in the supplement. I think the paper could be strengthened considerably with a major revision.

1) Some of the phenomena that are examined are classic, big effects that everyone knows, e.g. the face inversion effect, or Weber's law. Others are kind of obscure, drawn mainly from the authors' prior work (e.g. Experiment 7 and 8). Are all of them really necessary?

2) There are a lot of phenomena, and there is no justification for how they are chosen. Presumably there are other results that one could also test for. The "bag of results" feel gives the paper an atheoretical and ad hoc feel that I think will lessen its impact.

3) Some of the phenomena being tested are behavioral results in humans. Others are neural results. It would seem logical to probe for the behavioral phenomena by measuring the model's behavior, and the neural results by probing its internal representations, but that is not what is done. Instead, some behavioral phenomena are evaluated with measurements on the model's internal activations (e.g. Experiment 1 and 2, 10 and 11), whereas others are evaluated with the model's judgments (e.g. Experiment 3). This seems hard to justify, and potentially problematic for the cases where there are model-human differences. Of course, part of the benefit of a model is that one can easily probe the internals, and so that is a nice thing to do even for the behavioral phenomena, but I think it would be better to also measure behavior in some way, for instance by training classifiers on the activations to obtain the same judgment on the stimulus that a human makes in the experiment being considered.

4) The control results with the random weight network are important and need to be in the main figures. I was wondering about them throughout the paper and then finally found them in the supplement. They need to be part of the main paper. Same for the alternative networks. The fact that they show fairly consistent results greatly increases confidence in the overall conclusions, and so they should again be in the main figures.

Given 1, 2 and 4, one possibility might be to focus the paper more on a somewhat smaller set of well-

motivated phenomena, and to show the results more exhaustively for those (e.g. showing results for all the networks).

Minor comments:

Line 198 – conclusion seems the opposite of the actual result

Figure 4b – the correlation that is plotted could be made more explicit, e.g. by providing an equation

Line 321 – unclear – provide more details in main text

Line 340 – unclear – provide more details in main text

Experiment 9 – it would be more convincing to not rely just on this one stimulus to examine this issue

Line 412 and elsewhere – the writing is a little imprecise and colloquial for a scientific manuscript

Lines 455-457 – unclear

Lines 661-662 – I got confused by the fact that half the objects were rotated 90 degrees. Perhaps add to the figure and work through the logic a little more.

There are a few other relevant bits of literature that should be discussed, including the findings that adversarial examples fool humans in some cases, and that metamers of deep neural networks are far from metameric for humans.

Reviewer #2:

Remarks to the Author:

Review

First off, let me say, I'm very sorry for my delay—thank you for your patience.

In this manuscript, the representations learned in a standard feedforward convolutional deep neural network trained on object recognition were examined. Specifically, they considered 12 different signatures of human perception/macaque IT neurons, and examined whether these were emergent phenomena in the representations of the pre-trained networks. They found that some signatures were (e.g. thatcher effect, mirror confusion, Weber's law, relative size, etc), while others were not (e.g. relating to 3D shape, occlusion, and global advantage).

Overall, I think there is clear value in documenting the emergent capacities of object-recognition trained feed-forward neural networks. The logic of looking for emergent representational signatures in pre-trained networks, using stimuli and tasks which they were not trained on, is sound. Further, there is also inherent value in having one group conduct a number of these tests. Finally, the broad set of results are generally coherent and interesting. My main critiques are outlined below.

... Statement of findings

Throughout the results subsections, the main claims are all statements like this: "... deep networks show scene incongruence..." (line 198). "... deepnets show selectivity among multiple dimensions" (In 273). "... deepnets are not sensitive to 3d-shape" and "deep nets do not understand occlusions", etc. These statements are too general to me, implying relationships beyond what you have shown with your experiments.

First, to make these statements, for each experiment, it would be important to show that the same things hold over architectures, and over distance metrics (see comment below). I know some of this information can be figured out by scrutinizing the supplementary figures, but I would appreciate, for each case, a note about the extent of generality. If it's not true for all architectures and/or distance metrics, the claim could be more like "deep nets can show ...". For me, any existence proof is sufficient for the emergent point, and I value knowing whether the human-like representational signature is highly robust or more variable across these architectural/analytical variations.

Second, it is also the case you're testing one specific pre-trained VGG network. You have not shown that this holds over other VGG-16 instances, trained with a different random seed. Either qualify the result and highlight this as a future point. Or, I would be satisfied with replicating the results in one other VGG-16 instance (like monkey studies, $n=2$).

Third, there are also nuances to the pattern of results that are never acknowledged, e.g. some signatures emerge in conv blocks and others only in fully-connected. Some signatures are present in random networks and others are not. I see that discussing layer-wise effects is beyond how you've scoped your paper, which is fine with me. However, it should be accompanied with a little more acknowledgement of this up in the intro. For example, saying something pre-emptively like: "Here, we will consider the pre-trained deep network as showing the signature if any one layer shows the phenomena, and will highlight open questions for understanding the layers at which these signatures happen. Further, it would also help to revisit this in the discussion, to clarify further work to be done understanding differences in layer-wise progression of these signatures.

Fourth, for clarity, I think you should set up explicitly what you mean when you say "deep nets" in the intro. Something like, "here we will make claims about "deepnets", specifically referring to feedforward, deep convolutional neural networks, that have been trained to do object recognition using the Imagenet challenge." Better still, maybe be more specific with what you'll call them throughout, e.g. "Object-recognition deepnets"

... Framing

The paper narrative is framed from the point of view that there is "wide belief" that deep nets see like humans, and can be fixed with minor changes to data or architecture (e.g. line 13, 30, 49, 505, 516). To me, I think there is wide belief that this question is a highly debated issue: The glass-half full people hold the stance that deepnets are pretty good models (the best to date) but still need some architectural updates (e.g. as indicated by the push in current approaches to add recurrence, make shallower networks, etc.). The glass-half empty people note the clear mismatches between these models, e.g. the now widely known finding that deepnets overemphasize texture rather than shape (Geirhos et al., which you also find, and cite).

The point here is that situating the question more as "in what ways do deepnets see like us and in what ways don't they?" might hit your interested audience a little more square on, rather than (e.g. for me) having a knee-jerk "no we don't think that!" reaction from the get go. This would also let you cite the prior literature in the intro that have looked at some of these signatures, e.g. Geirhos and shape, Kriegeskorte and occlusion, etc, without undermining your motivating question.

... Methods

One of the key methods used in most of the experiments is a ratio of distances, where zero is the null hypothesis, and perceptual signatures show a bias one way or the other. My overarching question is: To what extent to these patterns of results you find depend on the Euclidean distance metric choice you used for computing deep-net activation differences. A standard in the fMRI literature is correlation distance ($1-r$), for better or worse, and the users of representational similarity methods are currently grappling with the question of which distance metric to use and why. Because this is still an open theoretical quagmire, I think it is important to repeat your key analyses with a correlation distance for the supplement (maybe Spearman, since using rank order is perhaps the most different from using

Euclidean, but I'd be interested in Pearson too, either way). Then, discuss whether the conclusions are the same or different, and if there are difference general conclusions you'd draw, make a case for why you favor euclidean distance from a theoretical standpoint.

... Comparison with humans / neurons

I found myself uncertain about the quantitative comparisons between the magnitude of deep-net biases and the magnitude of perceptual / neural effects. There are a lot of assumptions going into this, e.g. I recall that in one of the studies, the stimuli weren't even the same, others need normalizing... And, with 12 experiments, it's a lot to go through each method to figure out what assumptions you made when relating these. You probably have a sense of which ones were a little more of a stretch, and which one's match really well—my instinct is to just be a little more up front about that in each results sub-section, in a case-by-case basis, for the reader to situate the findings. Safer still, another approach might be to purely claim qualitative relationships, and provide the behavioral/neural data for quantitative context given linking assumptions.

... Integrative points

Given the multi-experiment structure of the paper, and depending on how much work you want to do, I would find it very valuable if you tried to situate all the results together. For example, there could be a very valuable and informative summary plot that highlights the 12 signatures and the layers the phenomena emerges. This would help highlight open questions about layer-wise affects (which you aren't addressing here), and would further would help you highlight that some of the 12 effects you tested are perceptual signatures from human behavior, some are neural signatures in single units, and some are neural population level signatures.

...Thatcher effect:

I think you need to revise your claim a little on this section: "the thatcher effect is present only in deep networks trained on upright recognition but not on generic object recognition." If you hadn't done the face-specific network, my guess is that you would have claimed that "deepnets show the thatcher effect, to a weak extent" because at least one of the layers in the object-trained network is positive. Narrative suggestion: Given that every other experiment is on object-trained deepnets and uses that one-layer-equals-success logic of the conclusions, you could adjust this results subsection. First, focus on the object trained one, making the claim that it can, but only weakly. Then, saying for comparison you did a face-trained deepnet and it does shows the effect even more. Thus, the claim could be changed to "object-recognition deepnets weakly show the thatcher effect, but this effect is substantially amplified on networks trained to recognize faces."

... Data visualization

I very much appreciate that you have a schematic of each experiment and the results. I also am very glad you show all the stages of the layer (indicated with a different marker type. Cool!). And, I very much appreciate that you highlight the comparison is with perception, or IT responses. However, I had trouble comparing across plots (e.g in the main manuscript and the supplement especially). Some suggestions: Make the zero point always in the middle of the y-axis, with an equivalent amount of axis in the positive and negative direction (when the effect could go either direction). Then, along the y-axis, always make the "human-like" direction be up (if possible, I think you already do this), and indicate that with an arrow annotation / label nearer to the axis itself. Right now you have that human-like area covered in gray, but that part of the graph almost seems grayed out, and harder to see the lines (maybe lighter gray?). By having the zero-line always in the middle, it's very orienting to understand the and compare across plots. I know this means there may be a bit of white space, but I think that white space is valuable in the context of considering all the effects together, and which go up and which go down.

... other points:

I found the Experiment 5 results subsection difficult to follow. I was not sure how to interpret the plots (correlation over what? Had to dig into the methods to understand.) I wondered about links to about

weber fractions and weber coefficients in Experiment 6. I found the Manipulation in Experiment 7 hard to follow, and couldn't easily decipher what parts to focus on for the relative change in the depicted figure.

... General discussion

line 575: "comparing human vision and deepnets depends on architecture, learning algorithm, and dataset and task" \diamond how does this relate to your general claims about "how deepnets see"? The challenge is definitely that each experiment could have its own discussion separately, and you made relatively few integrative statements about all of these signatures. There's a breadth-vs-depth challenge to the discussion, I found myself wanting to know more of your thoughts on the nuances in the patterns of the data (why do some signatures appear sooner vs later). I could see a whole results section, (or different paper?), highlighting the key results of the random network findings, and discussing the implications of how much of how deepnets see is related to the architecture or the learning process.

Response to reviewer comments

We thank the editor and reviewers for their careful reading and constructive comments. We have now done our best to respond to them fully through additional analyses and text revisions. Our responses are given below in red and refer to the appropriate points in the revised manuscript.

Reviewer #1:

Review of Jacob et al.

I apologize to the authors and editor for taking a long time to review this paper.

This paper evaluates similarities between biological visual systems and artificial neural networks trained to recognize objects. Artificial networks have recently become favored models for ventral visual stream processing, based on their ability to predict neural responses and replicate patterns of behavioral errors. This paper provides further tests of these similarities by assessing twelve different phenomena documented in the behavior or neural responses of primate visual systems. With a few exceptions, the general methodology is to measure distances between the activations evoked by pairs of images that are perceived as more or less similar by humans. The findings are that some of the twelve phenomena are evident in the artificial neural networks while others are not. The authors argue that these reveal some qualitative differences with biological vision.

Overall assessment:

This paper has some interesting results. There is a lot of interest right now in comparisons between artificial neural networks and biological sensory systems, and these results will find an audience because of that.

- We are glad that you found our study interesting.

The weaknesses of the paper are that 1) some of the phenomena seem more interesting/important than others,

- We agree with your observation but it is difficult to predict in advance which of these phenomena would prove to be useful in bridging the gap between human and deep network performance. Therefore we sought to capture all possible perceptual and neural phenomena that could be captured in terms of representational distances and therefore compared between brains and deep neural networks. We have now clarified this in the introduction (p. 5).

2) the phenomena are not obviously chosen in a principled way, and as a result the comparisons seem fairly disconnected from any sort of theory,

- We share your concern. Since there are no overarching theories for vision or deep neural networks, we can only characterize a variety of phenomena with the hope that some of these can yield deeper insights. We have now clarified this in the Introduction (p. 5).

3) the human and model measurements are not made with a consistent methodology, • We don't quite understand what you mean. In nearly all experiments, we have compared perceptual or neural representations with models, and have even used the same images wherever possible. We have reworked the text to make this clearer.

4) many important control results are buried in the supplement. I think the paper could be strengthened considerably with a major revision.

• We have now moved many of the controls into the main text, and we hope the revised manuscript addresses all your concerns.

1) Some of the phenomena that are examined are classic, big effects that everyone knows, e.g. the face inversion effect, or Weber's law. Others are kind of obscure, drawn mainly from the authors' prior work (e.g. Experiment 7 and 8). Are all of them really necessary?

• We understand your concern, but it was not clear to us which effects would prove to be important constraints for deep networks and which would not. Therefore we selected as many neural and perceptual properties as possible that could be compared in deep networks without explicitly training them to show these properties. We have now clarified this in the Introduction (p. 5).

2) There are a lot of phenomena, and there is no justification for how they are chosen. Presumably there are other results that one could also test for. The "bag of results" feel gives the paper an atheoretical and ad hoc feel that I think will lessen its impact.

• You are absolutely right, but as stated above, we do not have widely accepted and testable theories of computations and representations in vision. In the absence of such theories, we believe it is vital to characterize this "bag of results" so that this could eventually lead to better theoretical accounts. We have now clarified this in the Introduction (p. 5).

3) Some of the phenomena being tested are behavioral results in humans. Others are neural results. It would seem logical to probe for the behavioral phenomena by measuring the model's behavior, and the neural results by probing its internal representations, but that is not what is done. Instead, some behavioral phenomena are evaluated with measurements on the model's internal activations (e.g. Experiment 1 and 2, 10 and 11), whereas others are evaluated with the model's judgments (e.g. Experiment 3). This seems hard to justify, and potentially problematic for the cases where there are model-human differences. Of course, part of the benefit of a model is that one can easily probe the internals, and so that is a nice thing to do even for the behavioral phenomena, but I think it would be better to also measure behavior in some way, for instance by training classifiers on the activations to obtain the same judgment on the stimulus that a human makes in the experiment being considered.

- Sorry if this was not clear. We have always compared neural or perceptual distances with model distances. In Experiment 1, the Thatcher effect has been measured using dissimilarity ratings in humans, and therefore is directly comparable with model distances. In Experiment 2, we calculated a mirror confusion index from previously reported neural distances, and are therefore directly comparable with model distances. In Experiment 10, the behavioural phenomena involve search times measured in humans. Since search times are longer when the target is similar to the distractors, we took the reciprocal of search times as a measure of dissimilarity. In Experiment 11, the human behaviour reported was again search times, and so we took the reciprocal search times as a measure of dissimilarity. We have clarified this throughout.

You are absolutely right in noting that Experiment 3 was the only one evaluated using classification accuracy. We did so because the scene incongruence effect reported in humans was a difference in classification accuracy, and we therefore calculated model accuracy for a direct comparison. However, prompted by your query, we performed a layer-wise analysis of scene incongruence using distance comparisons – these show that scene incongruence effects arise only in the last fully connected layers in all networks (Results, p. 11; Section S4).

4) The control results with the random weight network are important and need to be in the main figures. I was wondering about them throughout the paper and then finally found them in the supplement. They need to be part of the main paper. Same for the alternative networks. The fact that they show fairly consistent results greatly increases confidence in the overall conclusions, and so they should again be in the main figures.

- We have now included the random weight network in the main figures – thank you for this important suggestion! While we agree that the results for other networks are indeed reassuringly similar to the VGG-16 network reported in the main text, we felt it was best to leave these in the supplementary material for ease of exposition.

Given 1, 2 and 4, one possibility might be to focus the paper more on a somewhat smaller set of well-motivated phenomena, and to show the results more exhaustively for those (e.g. showing results for all the networks).

- As detailed above, we think it is vital to compare and characterize as many properties as possible between brains and deep networks since it is unclear which of them will prove to be important constraints. We have now clarified this in the Introduction (p. 5).

Minor comments:

Line 198 – conclusion seems the opposite of the actual result

- Thanks for catching this! We have fixed it now.

Figure 4b – the correlation that is plotted could be made more explicit, e.g. by providing an equation

- Thank you, we have now included the details in the Methods.

Line 321 – unclear – provide more details in main text

- We have reworked this text to make it clearer.

Line 340 – unclear – provide more details in main text

- We have reworked this text to make it clearer.

Experiment 9 – it would be more convincing to not rely just on this one stimulus to examine this issue

- We share your concern, but unfortunately the original human studies used very few stimuli (only 4 sets of shapes), and we had to use only these stimuli for the deep networks to keep the comparison fair.

Line 412 and elsewhere – the writing is a little imprecise and colloquial for a scientific manuscript

- We understand what you mean. We have rephrased the colloquial phrases now wherever possible.

Lines 455-457 – unclear

- We have reworked this text to make it clearer.

Lines 661-662 – I got confused by the fact that half the objects were rotated 90 degrees. Perhaps add to the figure and work through the logic a little more.

- Thank you, we have now reworked the text to make it clearer.

There are a few other relevant bits of literature that should be discussed, including the findings that adversarial examples fool humans in some cases, and that metamers of deep neural networks are far from metameric for humans.

- Thank you for the pointers. We have included them in the text now.

Reviewer #2:

First off, let me say, I'm very sorry for my delay—thank you for your patience.

In this manuscript, the representations learned in a standard feedforward convolutional deep neural network trained on object recognition were examined. Specifically, they considered 12 different signatures of human perception/macaque IT neurons and examined whether these were emergent phenomena in the representations of the pre-trained networks. They found that some signatures were (e.g. thatcher effect, mirror confusion, Weber's law, relative size, etc), while others were not (e.g. relating to 3D shape, occlusion, and global advantage).

Overall, I think there is clear value in documenting the emergent capacities of object-recognition trained feed-forward neural networks. The logic of looking for emergent representational signatures in pre-trained networks, using stimuli and tasks which they were not trained on, is sound. Further, there is also inherent value in having one

group conduct a number of these tests. Finally, the broad set of results are generally coherent and interesting. My main critiques are outlined below.

- We are glad that you found our study interesting and we hope that the revised manuscript addresses your concerns.

... Statement of findings

Throughout the results subsections, the main claims are all statements like this: "... deep networks show scene incongruence..." (line 198). "... deepnets show selectivity among multiple dimensions" (ln 273). "... deepnets are not sensitive to 3d-shape" and "deep nets do not understand occlusions", etc. These statements are too general to me, implying relationships beyond what you have shown with your experiments.

- We now show that our results generalize to other instances of the VGG-16 networks (Section S1), other deep networks (Section S2) and using other distance metrics (Section S3).

First, to make these statements, for each experiment, it would be important to show that the same things hold over architectures, and over distance metrics (see comment below). I know some of this information can be figured out by scrutinizing the supplementary figures, but I would appreciate, for each case, a note about the extent of generality. If it's not true for all architectures and/or distance metrics, the claim could be more like "deep nets can show ...". For me, any existence proof is sufficient for the emergent point, and I value knowing whether the human-like representational signature is highly robust or more variable across these architectural/analytical variations.

- Thank you for this important comment. We have now included additional analyses on several other deep networks as well as another VGG-16 network initialized with a different random seed, as well as our results using several other distance metrics (Section S1-3). We found reassuringly similar trends as reported in for the main VGG-16 network. Based on this we feel confident in making the more general conclusion that deep networks (at least the ones trained on object classification) do or not have the effect we are investigating.

Second, it is also the case you're testing one specific pre-trained VGG network. You have not shown that this holds over other VGG-16 instances, trained with a different random seed. Either qualify the result or highlight this as a future point. Or, I would be satisfied with replicating the results in one other VGG-16 instance (like monkey studies, n=2).

- We now report similar results for another instance of VGG-16 (Section S2), which is trained with a different random seed and has a slightly lower top-1 accuracy on ImageNet.

Third, there are also nuances to the pattern of results that are never acknowledged, e.g. some signatures emerge in conv blocks and others only in fully-connected. Some signatures are present in random networks and others are not. I see that

discussing layer-wise effects is beyond how you've scoped your paper, which is fine with me. However, it should be accompanied with a little more acknowledgement of this up in the intro. For example, saying something pre-emptively like: "Here, we will consider the pre-trained deep network as showing the signature if any one layer shows the phenomena, and will highlight open questions for understanding the layers at which these signatures happen. Further, it would also help to revisit this in the discussion, to clarify further work to be done understanding differences in layer-wise progression of these signatures.

- Thank you for bringing up these nuances. We now acknowledge up front our criterion for when we deem that the network has a particular neural or perceptual phenomenon (Results, p. 7), and also acknowledge the nuances both in the Results and in the Discussion.

Fourth, for clarity, I think you should set up explicitly what you mean when you say "deep nets" in the intro. Something like, "here we will make claims about "deepnets", specifically referring to feedforward, deep convolutional neural networks, that have been trained to do object recognition using the Imagenet challenge." Better still, maybe be more specific with what you'll call them throughout, e.g. "Object-recognition deepnets"

- We now acknowledge this point explicitly in the beginning of the Results (p. 7).

...Framing

The paper narrative is framed from the point of view that there is "wide belief" that deep nets see like humans, and can be fixed with minor changes to data or architecture (e.g. line 13, 30, 49, 505, 516). To me, I think there is wide belief that this question is a highly debated issue: The glass-half full people hold the stance that deepnets are pretty good models (the best to date) but still need some architectural updates (e.g. as indicated by the push in current approaches to add recurrence, make shallower networks, etc.). The glass-half empty people note the clear mismatches between these models, e.g. the now widely known finding that deepnets overemphasize texture rather than shape (Geirhos et al., which you also find, and cite). The point here is that situating the question more as "in what ways do deepnets see like us and in what ways don't they?" might hit your interested audience a little more square on, rather than (e.g. for me) having a knee-jerk "no we don't think that!" reaction from the get go. This would also let you cite the prior literature in the intro that have looked at some of these signatures, e.g. Geirhos and shape, Kriegeskorte and occlusion, etc, without undermining your motivating question.

- Thank you for raising this issue, it is indeed important to sidestep this issue to bring focus on the key results directly. We have now reworked the text throughout accordingly.

... Methods

One of the key methods used in most of the experiments is a ratio of distances, where zero is the null hypothesis, and perceptual signatures show a bias one way or the other. My overarching question is: To what extent to these patterns of results you find depend on the Euclidean distance metric choice you used for computing deep-

net activation differences. A standard in the fMRI literature is correlation distance ($1-r$), for better or worse, and the users of representational similarity methods are currently grappling with the question of which distance metric to use and why. Because this is still an open theoretical quagmire, I think it is important to repeat your key analyses with a correlation distance for the supplement (maybe Spearman, since using rank order is perhaps the most different from using Euclidean, but I'd be interested in Pearson too, either way). Then, discuss whether the conclusions are the same or different, and if there are difference general conclusions you'd draw, make a case for why you favor euclidean distance from a theoretical standpoint.

• Thank you for raising this important point. We have now repeated all our analyses using other distance metric (cityblock, Pearson correlation, Spearman rank correlation) and found qualitatively similar results (Section S3).

... Comparison with humans / neurons

I found myself uncertain about the quantitative comparisons between the magnitude of deep-net biases and the magnitude of perceptual / neural effects. There are a lot of assumptions going into this, e.g. I recall that in one of the studies, the stimuli weren't even the same, others need normalizing... And, with 12 experiments, it's a lot to go through each method to figure out what assumptions you made when relating these. You probably have a sense of which ones were a little more of a stretch, and which one's match really well—my instinct is to just be a little more up front about that in each results sub-section, in a case-by-case basis, for the reader to situate the findings. Safer still, another approach might be to purely claim qualitative relationships, and provide the behavioral/neural data for quantitative context given linking assumptions.

• Thank you for bringing this up. We were unable to use matching stimuli for only 3 of the 12 experiments since we did not have access to them. We now acknowledge in the figure legend for each experiment whether the same stimuli were used, and have included these details clearly in the Methods.

... Integrative points

Given the multi-experiment structure of the paper and depending on how much work you want to do; I would find it very valuable if you tried to situate all the results together. For example, there could be a very valuable and informative summary plot that highlights the 12 signatures and the layers the phenomena emerge. This would help highlight open questions about layer-wise affects (which you aren't addressing here), and would further would help you highlight that some of the 12 effects you tested are perceptual signatures from human behavior, some are neural signatures in single units, and some are neural population level signatures.

• Thank you for this suggestion. We have now included a summary table showing the results for all networks and acknowledge these integrative points in the Discussion (p. 30).

...Thatcher effect:

I think you need to revise your claim a little on this section: “the thatcher effect is present only in deep networks trained on upright recognition but not on generic

object recognition.” If you hadn’t done the face-specific network, my guess is that you would have claimed that “deepnets show the thatcher effect, to a weak extent” because at least one of the layers in the object-trained network is positive. Narrative suggestion: Given that every other experiment is on object-trained deepnets and uses that one-layer-equals-success logic of the conclusions, you could adjust this results subsection. First, focus on the object trained one, making the claim that it can, but only weakly. Then, saying for comparison you did a face-trained deepnet and it does shows the effect even more. Thus, the claim could be changed to “object-recognition deepnets weakly show the thatcher effect, but this effect is substantially amplified on networks trained to recognize faces.”

- You’re right, thank you for pointing this out. We have now reworked our conclusion to reflect the above points (Results, p. 9).

... Data visualization

I very much appreciate that you have a schematic of each experiment and the results. I also am very glad you show all the stages of the layer (indicated with a different marker type. Cool!). And, I very much appreciate that you highlight the comparison is with perception, or IT responses. However, I had trouble comparing across plots (e.g in the main manuscript and the supplement especially). Some suggestions: Make the zero point always in the middle of the y-axis, with an equivalent amount of axis in the positive and negative direction (when the effect could go either direction). Then, along the y-axis, always make the “human-like” direction be up (if possible, I think you already do this), and indicate that with an arrow annotation / label nearer to the axis itself. Right now you have that human-like area covered in gray, but that part of the graph almost seems grayed out, and harder to see the lines (maybe lighter gray?). By having the zero-line always in the middle, it’s very orienting to understand the and compare across plots. I know this means there may be a bit of white space, but I think that white space is valuable in the context of considering all the effects together, and which go up and which go down.

- Thank you for this suggestion! We have now reworked all the plots to have the same y-axis limits and this looks much better now.

... other points:

I found the Experiment 5 results subsection difficult to follow. I was not sure how to interpret the plots (correlation over what? Had to dig into the methods to understand.)

I wondered about links to about weber fractions and weber coefficients in Experiment 6. I found the Manipulation in Experiment 7 hard to follow and couldn’t easily decipher what parts to focus on for the relative change in the depicted figure.

- We have now reworked the text in each case to make the text clearer.

... General discussion

line 575: “comparing human vision and deepnets depends on architecture, learning algorithm, and dataset and task” → how does this relate to your general claims about “how deepnets see”? The challenge is that each experiment could have it’s own discussion separately, and you made relatively few integrative statements about all

of these signatures. There's a breadth-vs-depth challenge to the discussion, I found myself wanting to know more of your thoughts on the nuances in the patterns of the data (why do some signatures appear sooner vs later). I could see a whole results section, (or different paper?), highlighting the key results of the random network findings, and discussing the implications of how much of how deepnets see is related to the architecture or the learning process.

- Thank you for these suggestions. We have now included these points in the Discussion.

Reviewers' Comments:

Reviewer #1:

Remarks to the Author:

Re-review of Jacob et al.

The revised paper is improved, and I support publication.

My comments are all minor.

There are assorted typos and grammar errors throughout the paper – it could benefit from a careful pass by a native English speaker.

The abstract is not very well written. Specifically, the description of the effects is in some cases too brief and vague to be interpretable.

The authors frequently describe their results as “interesting”, “insightful” etc. – I think it would be better to let the reader make this sort of evaluation for themselves, so I would suggest toning this down a bit.

Start of the results section – the distinction between “those that pertain to object or scene statistics” and “those that pertain to relational properties” was not completely clear to me – might not relational properties derive from scene statistics, e.g. scale invariance?

Line 158 – “see Methods” – just add the information to the main text – it will just take a sentence and will be easier on the reader

The Supplement is a little confusing as there are sections and figures that are both referred to as S1, S2 etc. I found the supplement very useful but the nomenclature could be cleaned up. Please also check all references to the supplement – some of them were to the wrong section.

Line 217 – “much larger degree” seems overstated – the effects look pretty comparable in the figure

Lines 304-306 – this conclusion is too general given that only a single architecture is described. Soften, or include results from some other architectures.

Line 371 – define tetrad

Line 596 – clarify that these are deep networks trained on a visual recognition task

Line 607 – “our” → “human”

Line 611 – “elucidating the necessary computations” – seems overstated and vague

Line 635 – typo

Line 726 – it would be appropriate to include a link to the architecture that was used, to enable reproducibility

Line 732 – “done” → “performed”

Line 740 – typo

Line 744 – typo

Line 746 – typo, “done” → “performed”

Line 771 – “Thatcher index” → “the Thatcher index”

Line 796 – “a few object” – give exact number

Line 811 – something got garbled here – rewrite

Response to Reviewers Comments

We thank the editor and reviewers for their continued careful reading and constructive comments that have greatly improved our manuscript. Our responses to the remaining concerns are in red.

Reviewer #1 (Remarks to the Author):

Re-review of Jacob et al.

The revised paper is improved, and I support publication.

- Thank you, we are delighted that you found our study interesting and insightful.

My comments are all minor.

There are assorted typos and grammar errors throughout the paper – it could benefit from a careful pass by a native English speaker.

- We have now proof-read our manuscript carefully for typos and grammatical errors. However, we are sad to see your unfounded comment that we are not native English speakers (we are) – this is both unnecessary and disrespectful.

The abstract is not very well written. Specifically, the description of the effects is in some cases too brief and vague to be interpretable.

- We have now rewritten our abstract to be more specific and to the point.

The authors frequently describe their results as “interesting”, “insightful” etc. – I think it would be better to let the reader make this sort of evaluation for themselves, so I would suggest toning this down a bit.

- We have now toned down these statements throughout.

Start of the results section – the distinction between “those that pertain to object or scene statistics” and “those that pertain to relational properties” was not completely clear to me – might not relational properties derive from scene statistics, e.g. scale invariance?

- By relational properties, we mean relations between objects. We have now clarified this in the text.

Line 158 – “see Methods” – just add the information to the main text – it will just take a sentence and will be easier on the reader

- We have now added a brief summary into the main text.

The Supplement is a little confusing as there are sections and figures that are both referred to as S1, S2 etc. I found the supplement very useful but the nomenclature could be cleaned up. Please also check all references to the supplement – some of them were to the wrong section.

- We have now checked all references to the supplement. Since references to the supplement are only to the relevant sections and not to the figures, we did not think it was too confusing to be numbering sections and figures this way.

Line 217 – “much larger degree” seems overstated – the effects look pretty comparable in the figure

- Thanks, we have now updated the text to reflect this.

Lines 304-306 – this conclusion is too general given that only a single architecture is described. Soften, or include results from some other architectures.

- We had already included results from other architectures in the Supplementary Material, and we have updated the text to reflect this.

Line 371 – define tetrad

- Thank you, we have now done so.

Line 596 – clarify that these are deep networks trained on a visual recognition task

- Thank you, we have now done so.

Line 607 – “our” → “human”

- We have now rephrased this sentence.

Line 611 – “elucidating the necessary computations” – seems overstated and vague

- True. We have now removed this phrase.

Line 635 – typo

- Thank you, fixed.

Line 726 – it would be appropriate to include a link to the architecture that was used, to enable reproducibility

- Thank you, we have added the link to the text.

Line 732 – “done” → “performed”

- Fixed

Line 740 – typo

- Fixed

Line 744 – typo

- Fixed

Line 746 – typo, “done” → “performed”

- Fixed

Line 771 – “Thatcher index” → “the Thatcher index”

- Fixed

Line 796 – “a few object” – give exact number

- We have now done so.

Line 811 – something got garbled here – rewrite

- Thank you for noticing this, we have now rewritten this part to make it clearer.